



# Identifying quasi-periodic variability using multivariate empirical mode decomposition: a case of the tropical Pacific

Lina Boljka[1], Nour-Eddine Omrani[1], and Noel S. Keenlyside[1]

[1]Geophysical Institute, University of Bergen and Bjerknes Centre for Climate Research, Bergen, Norway

**Correspondence:** Lina Boljka (lina.boljka@uib.no)

**Abstract.** Tropical Pacific is home to climate variability on different timescales, including El Niño Southern Oscillation (ENSO) – one of the most prominent quasi-periodic modes of variability in the Earth's climate system. It is a coupled atmosphere-ocean mode of variability with a 2-8-year-timescale and oscillates between a warm (El Niño) and a cold (La Niña) phase. However, the dynamics of ENSO is complex, involving a variety of spatial and temporal scales as well as their interactions, which are not necessarily well understood. We use a recently developed nonlinear and nonstationary multivariate timeseries analysis tool – multivariate empirical mode decomposition (MEMD) – to revisit quasi-periodic variability within ENSO. MEMD is a powerful tool for objectively identifying (intrinsic) timescales of variability within a given system. We apply it to reanalysis and observational data as well as to climate model output (NorCPM1). Observational/reanalysis data reveal a quasi-periodic variability in the tropical Pacific on timescales ∼2-4.5 years. This variability can then be related to ENSO's recharge-discharge and simplified West-Pacific oscillator conceptual models. The latter occurs only on this timescale and is not necessarily well represented in NorCPM1. Additionally, the ∼2-4.5-year variability in ENSO can be 'predicted' up to ∼20 months ahead, while predicting the full ENSO amplitude remains challenging. MEMD can therefore be used for assessing climate dynamics on different timescales and for evaluating their representation in climate models.

## 1 Introduction

The dynamics of the tropical Pacific is typically characterised by ocean-atmosphere interaction, whereby atmospheric changes in winds can lead to changes in the distribution of warm and cold waters in the ocean that in turn impacts the atmosphere. The variability in the tropical Pacific occurs on various timescales – from subseasonal to multidecadal (e.g., Maloney et al., 2008; Enfield and Mestas-Nuñez, 1999; Mestas-Nuñez and Enfield, 2001). One of the most prominent features in the tropical Pacific is the El Niño Southern Oscillation (ENSO), which is a quasi-periodic phenomenon occurring on (inter-annual) timescales of 2-8 years (e.g., Philander, 1990; Wang and Fiedler, 2006; Timmermann et al., 2018). ENSO events are characterized by warming sea surface temperatures (SSTs) during the development of El Niño (warm phase) and cooling of SSTs afterwards leading into La Niña (cold phase).

ENSO is associated with major changes in the precipitation distribution in the region (increased precipitation typically follows warmer SSTs; e.g., Wang and Fiedler 2006; Dai and Wigley 2000) as well as in the distribution of warm and cold waters in the upper ocean (warmer SSTs are associated with a deeper thermocline in the ocean). This has strong relevance



for the society (e.g., Wang and Fiedler, 2006; Santos, 2006; Lam et al., 2019) through modulating food production (fishery, agriculture), weather-related disasters, and economy (e.g., Cashin et al., 2017; Guimarães Nobre et al., 2019).

Because of its impacts, modelling and understanding of ENSO have been important topics for decades. The first models that captured crucial dynamics and were able to skilfully predict ENSO were developed in the 1980's (Quinn, 1974a, b; Cane et al., 1986; Zebiak and Cane, 1987; Suarez and Schopf, 1988), and the ENSO predictions have been continuously improving since (L'Heureux et al., 2020). To improve (long-term) predictions of ENSO and its impacts, it is important to study ENSO variability and associated physics. Indeed, ENSO can be described by several different processes and conceptual models that can yield/explain prediction skill on longer timescales.

First, one can use conceptual models based on red noise arguments, where relatively slow ocean variability is an 'integral' response to stochastic higher-frequency atmospheric (e.g., "weather") variability (following, e.g., Hasselmann, 1976; Frankignoul and Hasselmann, 1977). While this is an integral part of the dynamical coupling between atmosphere and ocean, Clement et al. (2011) argued that the thermodynamic air-sea coupling (via thermally coupled Walker mode) also matters. Also, atmospheric 'wind-forcing' can help triggering ENSO events, however the timescale of this 'forcing' is still debated (e.g., Roulston and Neelin, 2000; Capotondi et al., 2018) and its efficiency in triggering ENSO may depend on the background state (e.g., Lopez et al., 2013; Lopez and Kirtman, 2014; Timmermann et al., 2018). Nonetheless, these processes give the system some persistence and thus can be related to prediction skill on longer timescales.

Second, as ENSO is a quasi-periodic event (e.g., Wang et al., 2017) a part of its prediction skill likely comes from its internal quasi-periodic variability (e.g., Ghil and Jiang, 1998). Thus, ENSO's quasi-oscillatory dynamics has been studied extensively in the past, and the theories typically consist of positive (e.g., Bjerknes) and negative feedbacks between the atmosphere and ocean. The Bjerknes feedback (Bjerknes, 1969) refers to any decrease (increase) of trade winds that leads to reduced (enhanced) ocean upwelling (downwelling) and thus warming (cooling) in the eastern tropical Pacific leading to reduced (enhanced) zonal SST- and pressure-gradients, which in turn reinforce the initial increase (decrease) of the trade winds. This feedback alone would result in continuous warming (cooling) in the eastern tropical Pacific, therefore negative feedbacks are necessary for quasi-oscillatory behaviour in the eastern tropical Pacific (e.g., Wang et al., 2017).

To describe the interplay between the (Bjerknes) positive and negative feedbacks, several conceptual oscillator models have been proposed (e.g., Wang, 2018, see also section 3.2): (i) the delayed oscillator (e.g., Suarez and Schopf, 1988; Battisti and Hirst, 1989) that included reflected Kelvin waves at the western ocean boundary as a negative feedback; (ii) the recharge-discharge oscillator (e.g., Jin, 1997a, b; Burgers et al., 2005) that included Sverdrup transport as a discharge/recharge processes (negative feedback); (iii) the advective-reflective oscillator (e.g., Picaut et al., 1997; Wang, 2001a) that included advection and reflection of Rossby waves at the eastern ocean boundary as a negative feedback; and (iv) the Western Pacific oscillator (e.g., Weisberg and Wang, 1997; Wang et al., 1999) that included interactions with the West-Pacific wind-forced Kelvin waves as a negative feedback. These oscillators can all be viewed as part of the unified oscillator (Wang, 2001a, 2018, see also section 3.2).

The above oscillators describe different processes that can lead to an ENSO event. These processes generally involve changes in the thermocline depth, surface wind stress, and SST anomalies. However, ENSO has many flavours and can, for example,





occur in the Eastern or Central Pacific (EP and CP ENSO, respectively; e.g., Kao and Yu 2009; Singh and Delcroix 2013; Zhang et al. 2019), and can also occur on different timescales, e.g.: (i) a quasi-biennial (QB) ENSO with a ∼2 year timescale and (ii) a low-frequency/quasi-quadrennial (LF/QQ) ENSO with a ∼4-year timescale (e.g., Jiang et al., 1995; Allan, 2000; Kim et al., 2003; Keenlyside et al., 2007; Bejarano and Jin, 2008; Jajcay et al., 2018; Froyland et al., 2021).

Due to a multitude of factors impacting ENSO (described above), its predictability remains challenging (e.g., Fedorov et al., 2003; L'Heureux et al., 2020). This is due to unpredictable atmospheric (stochastic) forcing, different local and remote factors that impact it (e.g., inter-basin interactions), and different spatial and temporal scales involved. All these processes impact the initial conditions, which ultimately determine the ENSO onset/magnitude and thus its prediction. Presently, we are able to predict ENSO reasonably well 6 months ahead, and in some (rare) special cases even up to 2 years ahead (e.g., Chen et al.,
2004; Park et al., 2018; L'Heureux et al., 2020). Some further improvements of ENSO predictability are also possible by utilising machine learning and neural network algorithms (e.g., Ham et al., 2019, 2021; Dijkstra et al., 2019). However, to improve and extend ENSO's (general) prediction range, further physical understanding of this phenomenon is needed, and with it also model improvements (e.g., McPhaden, 2015; L'Heureux et al., 2020).

The main aim of this study is therefore to explore the intrinsic variability of the tropical Pacific (and thus ENSO) on differ-
ent timescales, by objectively splitting the tropical Pacific variability (using reanalysis/observational products) into different timescales (identify intrinsic variability). This allows identification of potential oscillations and their timescales, as well as the physical mechanisms that contribute to the tropical Pacific (and thus ENSO) variability on these timescales (following, e.g., Jin 1997a; Wang 2001a). To achieve this, many different methods have been used in the past, e.g., multi-channel singular spectrum analysis (MSSA), principal oscillation patterns (POPs), linear inverse model (LIM) (Broomhead et al., 1987; Has-
selmann, 1988; Penland and Sardeshmukh, 1995; Ghil et al., 2002). However, these methods can be linear, stationary or both (Huang et al. 1998; Ghil et al. 2002; see also section 2.2), which can be a constraint for studying inherently nonlinear and non-stationary systems, such as the climate system.

Therefore, in this study we employ a recently developed nonlinear and nonstationary method for multivariate-timeseries filtering/analysis, called Multivariate Empirical Mode Decomposition (MEMD; Rehman and Mandic 2010). This method is a
multivariate extension of the Empirical Mode Decomposition (EMD; Huang et al. 1998) and can identify common timescales across different timeseries and multi-dimensional fields (for details see section 2.2 and Appendices A, B). This method objectively extracts intrinsic modes of variability in the tropical Pacific without a pre-selection of timescales, and can be used for identifying potential quasi-oscillatory behaviour as well as the physics related to the tropical Pacific (and ENSO) variability on different timescales (more in section 3, and Appendix B2). This knowledge can then be used for further understanding of
model biases, ENSO prediction, ENSO teleconnections across scales, and other processes (see section 4 and 5).

The manuscript is structured as follows. Section 2 provides data and methods used in this study (Appendices A and B provide further details on methodology); section 3 explores the physical mechanisms relevant on different timescales with a focus on the timescales that are quasi-oscillatory; section 4 compares climate model data with reanalyses/observations and tests predictability of the quasi-oscillatory mode. Conclusions are given in section 5.





## 2 Data and Methods

### 2.1 Data

In this study we focus on the representation of the ENSO using MEMD algorithm (described below). For this reason we analyse monthly mean data of four different variables: sea surface temperature (SST) from HadISST observational dataset (Rayner et al., 2003), surface zonal wind stress ($\tau_x$) and thermocline depth (i.e., the depth of the 20°C isotherm) both from SODA2 ocean-reanalysis dataset (Carton and Giese, 2008), and mean sea level pressure (MSLP) from NOAA20CR reanalysis dataset (Compo et al., 2011; Slivinski et al., 2019). The latter was used to assess ENSO teleconnections to the North Atlantic (i.e., ENSO impact on the North Atlantic Oscillation, NAO), which confirmed a signal consistent with previous work (not shown; e.g., Brönnimann 2007; Fereday et al. 2008; Jiménez-Esteve and Domeisen 2018; Hardiman et al. 2019; Jiménez-Esteve and Domeisen 2020). The former three are used to asses the ENSO dynamics as these quantities typically play a leading role in the onset and decay of ENSO events, and are typically used in the oscillator models that explain ENSO dynamics (see section 3.2 and, e.g., Wang, 2018). The data are analysed in the period 1871-2010 for which all datasets are available. Note that surface wind stress and the ocean subsurface data reconstructions in the $19^{th}$ and early $20^{th}$ century are less reliable than in the late 20th century due to sparser data coverage, thus the results presented here are only as accurate as these reconstructions can be (Wittenberg, 2004; Crespo et al., 2022).

The MEMD analysis (described below) is performed on all 4 fields simultaneously with SST at the highest resolution (1° in latitude and longitude), whereas thermocline depth, $\tau_x$ (both 9° resolution in longitude, 5° resolution in latitude), and MSLP (9° resolution in longitude, 6° resolution in latitude) have much lower resolution. This gives greater weight to SST data in the analysis, and less towards the other variables, such that the mode does not change significantly by adding other variables in the analysis (i.e., results below for the SSTs are similar whether we use SSTs alone or together with other fields). Note that MEMD can be sensitive to input data, thus we must carefully consider the input data structure (relevant to our study). The SSTs, $\tau_x$, and thermocline depth are computed in the tropical Pacific (110°E - 65°W, 25°S - 25°N), whereas MSLP (not shown) is computed in the North Atlantic sector (80°W - 30°E, 25°N - 70°N).

In Sect. 4 below we also analyse model data, namely the first ensemble member (i.e., r1i1p1f1; other ensemble members were qualitatively similar) of the NorCPM1 historical simulation (Bethke et al., 2019, 2021). In this study we only test the dynamics of ENSO in the model, thus only $\tau_x$, thermocline depth, and SSTs are used in the MEMD. For consistency, we use the time period 1871-2010 in the model as well.

While the SSTs, thermocline depth, and $\tau_x$ play an important role in the ENSO dynamics, it is specific regions (see Table 1) that are more relevant for the oscillator models (e.g., Wang, 2001a, 2018; Burgers et al., 2005). Namely, we average $\tau_x$ over Niño4 and Niño5 regions separately, thermocline depth over Niño6 region (off-equatorial thermocline depth) and over the tropical Pacific (Pacific mean), and SSTs over Niño3 region (see also Table 1 and Fig. 3 in Wang et al. 1999).

Before performing the analysis, we detrend the data and remove its seasonal cycle, by computing 30-year running mean seasonal cycle and subtracting it from the data at each grid point (no additional smoothing is performed) (similar to, e.g., de la





**Table 1.** Tropical Pacific regions used for computing timeseries (see text for details).

| Region | Latitude Range | Longitude Range |
|---|---|---|
| Niño3 | 5°S - 5°N | 150°W - 90°W |
| Niño4 | 5°S - 5°N | 160°E - 150°W |
| Niño5 | 5°S - 5°N | 120°E - 140°E |
| Niño6 | 8°N - 16°N | 140°E - 160°E |
| Pacific mean | 5°S - 5°N | 120°E - 90°W |

Cámara et al., 2019). This is done to avoid domination of the seasonal cycle or trend in the statistical analysis below, even though the method presented below can generally extract nonlinear trends by itself.

## 2.2 (Multivariate) Empirical Mode Decomposition

To analyse the data we use Multivariate Empirical Mode Decomposition (MEMD). This method was first introduced by Rehman and Mandic (2010) as a multivariate extension of the Empirical Mode Decomposition (EMD; also called Hilbert-Huang transform). The EMD was introduced by Huang et al. (1998) as an alternative method for time-filtering of the timeseries that is entirely data adaptive, nonlinear, and nonstationary, and thus more appropriate for analysing nonlinear and/or nonstationary data. This is in contrast to some other methods methods (e.g., singular spectrum analysis, wavelet analysis, Fourier transform, principal component analysis, nonlinear Laplacian spectral analysis, POPs, LIM), which do not necessarily have a multivariate extension, and/or are either linear, stationary, or both. The EMD is based on Hilbert transform and takes advantage of the instantaneous frequency, allowing a 'local' extraction of modes of variability.

The EMD method has a relatively simple implementation (Huang et al., 1998): (i) first we identify local minima and maxima of the timeseries and create an envelope by interpolating between the subsequent maxima (upper envelope) and between subsequent minima (lower envelope); (ii) then we obtain an average envelope from the upper and lower envelope and subtract it from the timeseries data; (iii) the subtracted data (i.e., original data minus the average envelope) become the first mode of variability with the signal of the highest frequency in the dataset, whereas the average envelope can be analysed further; (iv) repeat steps (i)-(iii) for the average envelope until only a trend (residual) remains, i.e., until a condition of at least 2 extrema in the dataset can no longer be satisfied. The modes of variability obtained through this process are called intrinsic mode functions (IMFs), and their instantaneous timescale is characterised by the time-lapse between two subsequent extrema (and IMF's mean timescale is an average over the instantaneous values). Each IMF also has to satisfy two criteria: (a) the number of extrema and the number of zero-crossings differs at most by one; and (b) the mean value of the envelope of the IMF is zero. Note that the procedure from (i) to (iv) does not necessarily satisfy (a)-(b) immediately, thus additional sifting process is used that requires a stopping criteria to ensure physical meaning of the IMFs. The stopping criteria can be based on the standard deviation of each



IMF, on the maximum number of iterations, etc., which set tolerance and confidence limits for the IMF (Huang et al., 1998; Rilling et al., 2003; Huang et al., 2003).

The MEMD method (Rehman and Mandic, 2010) is a generalisation of the EMD to multivariate datasets of more than two timeseries (for bivariate and trivariate data separate methods exist; Rilling et al. 2007; Rehman and Mandic 2010).
The method solves a similar problem as in (i)-(iv) but on an N-Sphere, and also retains similar stopping criteria for the sifting process. For further details on the method the reader is referred to Rehman and Mandic (2010). The code for the method is freely available on Github (https://github.com/mariogrune/MEMD-Python-; similarly for the EMD discussed above: https://github.com/laszukdawid/PyEMD), and the user ultimately only decides about the stopping criteria, which are here set to their "fix_h" parameter, following Huang et al. (2003) who suggest that limiting iterations yields better-behaved IMFs than
other stopping criteria. Here, we limit the number of iterations to 15 (parameter "n_iter" is 15), though other values were tested and a range for "n_iter" around 10-30 yielded similar results, suggesting some convergence for the significant modes of variability. Note that at higher frequencies we find mode-mixing in our MEMD analysis (where timescales are not clear), especially with larger number of iterations, however the significant modes of variability on interannual timescales that are of interest here are largely unaffected by this.

The MEMD ultimately extracts timescales common to all input timeseries and provides IMFs consistent with these timescales for each input timeseries. The use of such IMFs will depend on application—here we analyse principal components of a 3-D field to extract quasi-oscillatory modes of variability (the whole procedure is described below and in Appendices A, B).

Here, we use the MEMD method because of its nonlinear and nonstationary properties, which allows a detection of quasi-periodic signals without pre-selecting (filtering for) a frequency band (a constraint with, e.g., singular-spectrum analysis), and
170 without any periodic signal or basis of functions specifications (a problem with, e.g., Fourier transform and wavelet analysis). Even though EMD and its 1D extension Ensemble EMD (EEMD; Wu and Huang 2009) have been applied in climate science in various applications, e.g., for smoothing, filtering, extracting trends, variability, and testing for red noise distribution of climate data (e.g., Duffy, 2004; Wu et al., 2007; Franzke, 2009; Lee and Ouarda, 2011; Qian et al., 2011; Franzke and Woollings, 2011; Franzke, 2012; Ezer and Corlett, 2012; Ezer et al., 2013; Wang and Ren, 2020), it has not been explicitly used for extracting
quasi-periodic signals. Moreover, the MEMD has only been applied to an analysis of the atmosphere-ocean coupling strength (Alberti et al., 2021) in climate science, which was done in a more idealised setting from the present study. Therefore, we also perform extensive analysis of the method itself and compare it to the basic band-pass filtering (5th order Butterworth filter) and to Fourier transform analysis, to show that its results are consistent with other methods, but can also extract more information in an objective way (see below and Appendices A, B).

As with all statistical methods, also (M)EMD has drawbacks that we must be aware of (e.g., Stallone et al., 2020). Because it is a statistical method we must always test if the modes we find using (M)EMD are physical, and also check for convergence (stability) of the modes using different parameter sweeps (as mentioned above). Much like other timeseries-filtering methods, also (M)EMD has issues at the edges of the timeseries, which can lead to "travelling waves" and thus unrealistic peaks in the timeseries (e.g., Stallone et al., 2020). There is also a common issue of mixing modes (as mentioned above) where one
(real) mode is split into two or more modes if we choose "wrong" parameters (e.g., Huang et al., 1999, 2003; Stallone et al.,





2020), but this is also an issue when there is no clear timescale-separation. We have tested different sweeps of parameters and have ultimately decided on the ones specified above, as they suggested some stability and the results were realistic and comparable to other methods (note that for other applications different parameter sweeps may yield better results). Therefore, we continue from hereon with the description of implementation of the MEMD method and later we use it to understand the

ENSO dynamics on different timescales.

**Obtaining modes of variability via MEMD**

As mentioned in Sect. 2.1, we use 3-D data, i.e., $\mathbf{A}(t,y,x)$ (with $t$ as time, $y$ as latitude, $x$ as longitude, $\mathbf{A}$ as the selected variable(s)/field(s); bold letters represent matrices), as basis for the MEMD analysis. We first remove 30-year running mean seasonal cycle (de la Cámara et al., 2019) from $\mathbf{A}$ to get a temporal anomaly, $\mathbf{A}'$. If we use more than one variable (e.g.,

SSTs, surface wind stress, thermocline depth, MSLP) in the analysis, we concatenate the different variables in their spatial dimensions, i.e., we get $\mathbf{A}'(t,y=y_1+y_2+...,x=x_1+x_2+...)$ (with subscripts 1,2,... representing spatial dimensions of the different variables). Note that we also divide data by their standard deviations before concatenating them. Then we reduce the dimensionality by computing spatial patterns (empirical orthogonal functions, EOFs) and their timeseries (principal components, PCs) via singular value decomposition (SVD). We only retain the first 20 PCs that explain the majority of the variance

in the field $\mathbf{A}'$. The PCs are then ultimately used as input data for the MEMD algorithm as described in Appendix A in detail. The modes that emerge from the MEMD analysis can become spatio-temporal modes of variability (IMFs) by reconstructing the field from PCs and EOFs (Appendix A).

To compute an index, such as Niño3, we can average over a x-y region (Table 1) from $\mathbf{A}'$ to obtain timeseries for each IMF separately. This gives the contribution of the intrinsic variability mode components (from IMFs) to the index.

Once we have computed the IMFs from the input data, we need to test if they are significant. The importance of each IMF can be assessed by computing variance explained of each IMF relative to the input field (e.g., retaining those IMFs that explain more than 0.1% variance) or through other significance tests (e.g., white noise test; Appendix B1; Wu and Huang, 2004). However, as the interest in this study is potential oscillatory behaviour in the ENSO region, we seek IMFs that pass a red noise threshold as the monthly sea surface temperature data typically follow a red noise distribution. The red noise test is described

in Appendix B2, and the results are shown in Fig. B1 (incl. a white noise test), which is further confirmed via a typical power spectrum (via Fourier Transform) analysis in Fig. B3.

The results from Appendix B2 suggest that there are one to two significant modes of variability in the ENSO region. Note that any potential oscillatory timescales longer than 30-years were removed via 30-year running mean mentioned above, and are thus not considered here. The significant modes of variability in the ENSO region have average timescales of about 23 and

37 months (i.e., 2-3 years) and are therefore well within the typical ENSO timescale range (2-8 years). Such timescales (with their uncertainty ranges) are consistent with both QB and LF/QQ ENSO (e.g., Jiang et al., 1995; Jajcay et al., 2018). A clearer identification of the quasi-periodic mode of variability for the mode with 37 months average period is also consistent with 'more periodic' LF/QQ ENSO (Jiang et al., 1995). Spatial pattern of the significant mode with 37 months timescale is provided in Fig. 2, which shows that it indeed resembles ENSO spatial pattern, and is further discussed in section 3. Note that the spatial





pattern of the 23 months mode looks qualitatively similar but it is less significant, hence it is not shown for brevity. On longer timescales we do not find any behaviour that would be discernible from red noise, suggesting that the lower-frequency range of ENSO (timescales longer than ∼4.5 years) (Allan, 2000; Jajcay et al., 2018, e.g.,) is better represented by red noise and likely less predictable than QB or LF/QQ ENSO.

Even though the results from Appendix B2 give us 'significant' IMFs, MEMD remains a statistical method and thus further analysis is required to identify whether its modes have any physical meaning. If IMFs are physical, such information could be used for improving our climate models, long-term prediction, understanding the teleconnections, (as mentioned above) the underlying physics/variability of the field we are interested in (e.g., ENSO), etc. This is addressed in the following sections (3, 4).

## 3 Dynamics of ENSO inferred from MEMD

In the previous section we have established that the Niño3 index in the tropical Pacific exhibits two quasi-periodic modes with average periods ∼2-3 years (via MEMD analysis; note that from hereon we only consider the results from the MEMD analysis). The timeseries of the two IMFs (IMF12 and IMF13) are shown in Fig. 1 (red dotted and dashed lines respectively), along with their sum (black dashed line) and a band pass (16-51 months) filtered Niño3 index (black solid line).

Fig. 1 shows that the period/frequency of the two modes is not constant (i.e., varies in time; see below), thus we also specify a range of periods/frequencies for the two modes. The mean periods of IMF12,13 with their 'uncertainty' ranges (in square brackets) are: 23 [16, 33] (IMF12), and 37 [26.5, 51] (IMF13) months. These ranges are defined based on the $6.7^{th}$ and $93.3^{rd}$ percentiles of IMF12,IMF13's instantaneous period/frequency values, which roughly correspond to $\pm1.5\sigma$ (instantaneous periods). Note that instantaneous period/frequency can be obtained via Hilbert transform (see Eq. (B4) in Appendix B1). This range was chosen as it yielded the best results, but other (reasonable) percentile ranges give qualitatively similar results. We then use these period/frequency ranges to perform a band pass filter (via $5^{th}$ order Butterworth filter) for the individual modes (e.g., in Fig. 3b), as well as for the sum of the modes where the band-pass range encompasses periods of both modes, i.e., 16-51 months (e.g., in Fig. 1).

Timeseries of the sum of IMFs (black dashed line) and band passed index (black solid line) in Fig. 1 largely agree, i.e., their correlation is 0.96. However, also the individual modes show very good agreement with the band-passed index with correlations of 0.71 (IMF12) and 0.81 (IMF13), which can be increased further if we consider only the specific IMF's timescale range (specified above) when band-passing the Niño3 index. This merely confirms that MEMD results are consistent with other filtering methods.

Note that the timeseries of Niño3 index extracted from IMF12,13 (Fig. 5) are largely consistent with the modes identified in Jiang et al. (1995) (their Fig. 9a), who used MSSA analysis, and with Wang and Ren (2020) (their Fig. 3), who used EEMD on Niño3.4 index. IMF13 is also similar to Froyland et al. (2021) 4-year mode (their Fig. 10), who used operator-theoretic approach. Also, the average periods of the 'significant' modes of variability (IMF12,13) in this study are typically lower than in other studies, however please recall that we used much longer timeseries, i.e., within the overlapping time periods the timescale





(and corresponding timeseries) is generally consistent across different studies. These similarities provide further confidence in the results provided below. Note also that these studies have focused on different ENSO timescales and associated different

spatial patterns during QB and LF/QQ ENSO events, but have not considered the conceptual models that we analyse below.

The period-ranges provided above also suggest that there is some overlap between the period/frequency bands, which is a result of nonlinear and nonstationary evolution of the modes (period is not constant as seen in Fig. 1). Indeed, there is a low (yet significant) correlation between the two modes ($\sim$0.27). Thus, in some time-windows the two IMFs can describe the variability of a similar timescale (e.g., similar time evolution in Fig. 1 around years 1902-3, 1923-7, 1983, 1997), but in other

time-windows they describe variability on different timescales. Despite changing periods, the two IMFs overall pass the red noise threshold and are therefore still considered significant and thus *quasi*-periodic (Appendix B2).

Fig. 1 also shows that in some decades the band-pass-filtered Niño3 index (solid black line) is more consistent with the lower frequency IMF13 (red dashed line; approx. 1870-1917, 1938-1950, 1968-2000) and in other periods with the higher frequency IMF12 (red dotted line; approx. 1917-1938, 1950-1968, 2000-2010). This is likely consistent with the interdecadal shifts in the

265 frequency of ENSO (Hu et al., 2017, 2020) that have occurred around years 1970 (from higher frequency to lower frequency) and 2000 (from lower frequency to higher frequency), and can also be seen earlier in the record in Fig. 1. We can also see interdecadal changes in amplitude of ENSO modes, i.e., middle panel (periods 1920-1965) in Figs. 1, 5 compared with top and bottom panels (periods 1870-1920, 1965-2010). This is somewhat consistent with Crespo et al. (2022), who found reduced amplitude of ENSO during 1901-1931 and 1935-1965 periods relative to post-1970 period.

Note that the magnitude of ENSO ultimately depends on all underlying modes of variability in the tropical Pacific (not just on the IMFs discussed here). In fact, we find $\sim$4 modes with average timescales ranging from $\sim$1.5 to $\sim$15 years that can reproduce the majority of ENSO variability (not shown), but the rest of these modes are consistent with red noise.

Fig. 1 also shows that weak Niño3 events have either small amplitudes (e.g., 1933-7) of both IMFs or opposite amplitudes (e.g., 1908-11), whereas strong Niño3 events generally show a constructive interference or mode-combination (e.g., 1997, a

275 super-El Niño event), which is consistent with, e.g., Slawinska and Giannakis (2017); Jajcay et al. (2018); Wang and Ren (2020); Froyland et al. (2021).

Note that the band-pass range here was chosen using an 'objective' data-driven approach, which shows that MEMD can be used as an effective band-pass filter method. This is supported by a good agreement between the band-passed data and the IMFs (similarly, with spectral analysis; Appendix B2.2; Fig. B3), which suggests consistency between modes obtained via MEMD

and other statistical methods. However, we must still test whether the obtained IMFs are physical, thus we now turn to ENSO dynamics to address this question.

## 3.1 Observational evidence

ENSO evolution is primarily related to the evolution of the ocean surface wind stress ($\tau_x$), thermocline depth and the SSTs in the tropical Pacific (e.g., Wang, 2018). Their relationship is demonstrated with map phase composites of IMF13 in Fig. 2,

where shading represents SST anomalies, contours represent the thermocline depth anomalies (solid contours represent positive



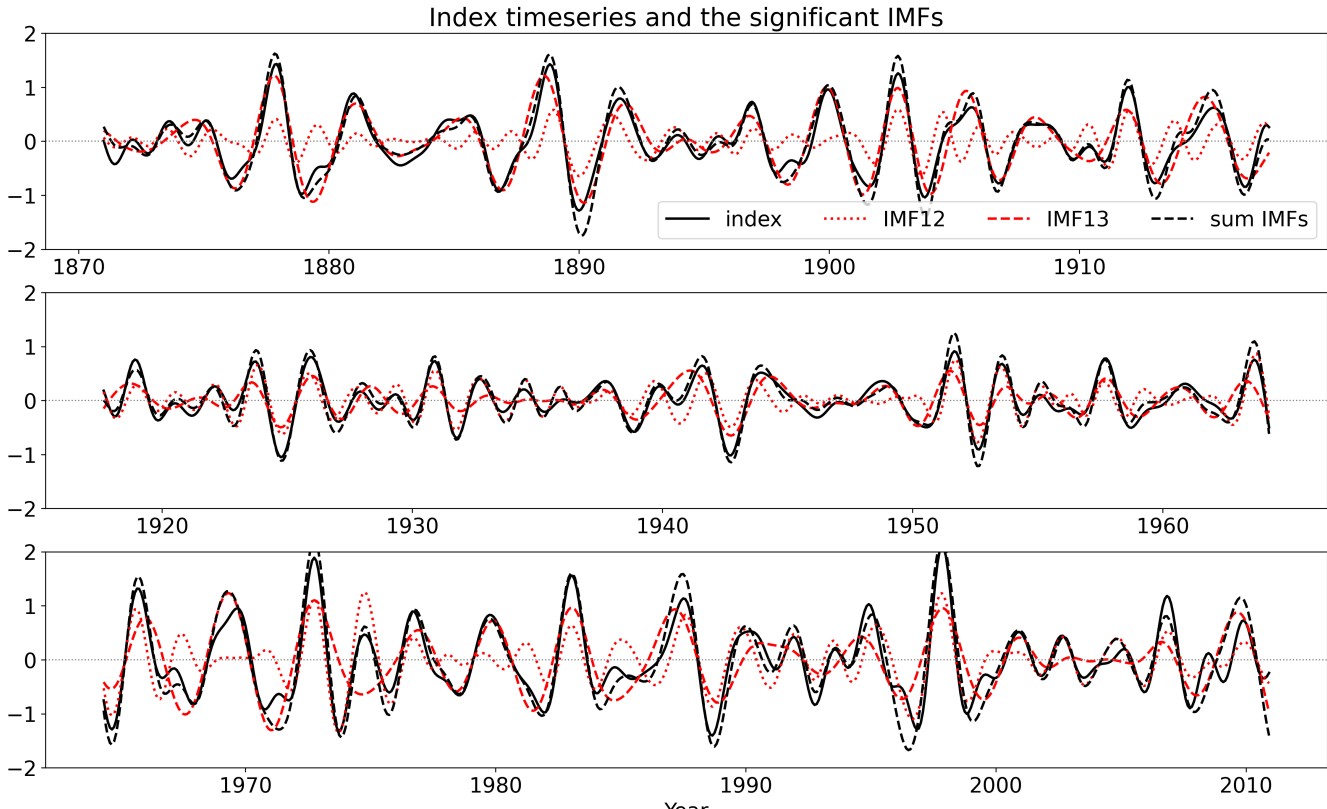

**Figure 1.** Timeseries (1871-2010) of band-pass filtered (16-51 months) Niño3 index (black solid line), IMF12 obtained via MEMD (red dotted line), IMF13 obtained via MEMD (red dashed line), and the sum of the two IMFs (black dahsed line).

values and deeper thermocline), yellow arrows represent $\tau_x$ anomalies, and grey shading represents grid points of the SSTs that do not pass the red noise threshold. Note that all values are standardised (i.e., divided by standard deviation).

The phase composites are computed using instantaneous phase of the PC1 (of IMF13) timeseries that we can obtain via Hilbert Transform (Appendix B1, Eq. (B4)). This 'assigns' every point in the PC1 timeseries a phase between 0 and 360

degrees, which can then be split into 12 phases (marked phase 0 through 11) and all points in timeseries (of 1-D or 3-D fields) belonging to a specific phase are then averaged to form phase composites shown in Fig. 2. Other phase composites below are constructed similarly but the timeseries they are composited over is typically SST (Niño3) instead of PC1 (as specified in figure captions; e.g., Fig. 3), as conceptual models discussed below (section 3.2) involve SST (Niño 3). Nonetheless, spatial composites (Fig. 2) still reflect the relevant physics (discussed below) and are consistent with the SST (Niño3) composites

(Fig. 3). Note that PC1 is used for spatial composites as it reflects the variability of full spatial field, not just SST (Niño3), and their correlation is typically close to 1 (e.g., Ashok et al., 2007).



**Figure 2.** Latitude-longitude phase composite (phases 0 to 11 as labelled) of IMF13: shading for SSTs, contours for thermocline depth (contour interval is the same as in the colourbar with solid contours representing positive values, dashed contours negative values), and arrows for $\tau_x$ (the scale is shown in the bottom left corner of panels for phases 10,11). All data is standardised and all fields were composited based on the phase of the PC1 of the combined field.





Fig. 2 shows a typical cycle of ENSO in the tropical Pacific (on a ∼3-year timescale), which can be summarised also with line phase composites (Fig. 3a,c) of timeseries averaged over specific regions (as labeled; see also Table 1) of the tropical Pacific (e.g., Wang, 2018). Here we analyse SSTs (Niño3), $\tau_x$ (Niño4), $\tau_x$ (Niño5), thermocline depth (Niño6), and thermocline depth (Pacific mean).

Together the two figures (Figs. 2, 3a,c) suggest the following sequence of events: (i) during La Niña (phase 0) we have negative SST anomalies and shallower thermocline in Niño3 region, stronger easterly wind stress in Niño4 region, and deeper thermocline in Niño6 region; (ii) as La Niña weakens (phases 1-3), the westerly wind stress in Niño5 region and thermocline depth averaged across the tropical Pacific peak, starting the El Niño cycle; (iii) SSTs warm, thermocline (Niño3) becomes deeper, wind stress (Niño4) becomes westerly, and thermocline (Niño6) becomes shallower (phases 4-6); (iv) El Niño weakens (phases 7-9) and $\tau_x$ (Niño5) becomes easterly as well as the thermocline across the Pacific becomes shallower, starting a La Niña event (phases 10-11); (v) the cycle repeats. Note that there are also wind stress changes in the far eastern tropical Pacific that generally oppose the ones in the Niño4 region (Fig. 2), but occur roughly at the same time and are thus not explicitly considered further.

The evolution described above is also seen in the band-pass (26.5-51 months; 2-4.5 years) filtered data (Fig. 3b). Note that the values in Fig. 3b are slightly larger than in Fig. 3a, because slightly different frequency ranges are ultimately represented in the two panels, but they remain qualitatively similar. Similar results can also be obtained for the 16-33 months band-passed data and IMF12 (not shown). However, at lower and higher frequencies the evolution is different with $\tau_x$ (Niño5) closely following thermocline (Niño6) (Fig. 4a,b), whereas other variables remain similar across timescales (to ± 1 phase). This suggests a different behaviour of the high- and low-frequency tropical Pacific variability compared with the 1.5-4.5-year variability (IMF12,13) discussed here.

Since we observe clear relationships between the relevant variables (e.g., Fig. 2) that strongly resemble a unified oscillator of Wang (2001a) (see also Fig. 5 in Wang 2018), recharge-discharge oscillator (e.g., Burgers et al., 2005), and others, we now explore the theory of ENSO dynamics using the relevant conceptual oscillator models.

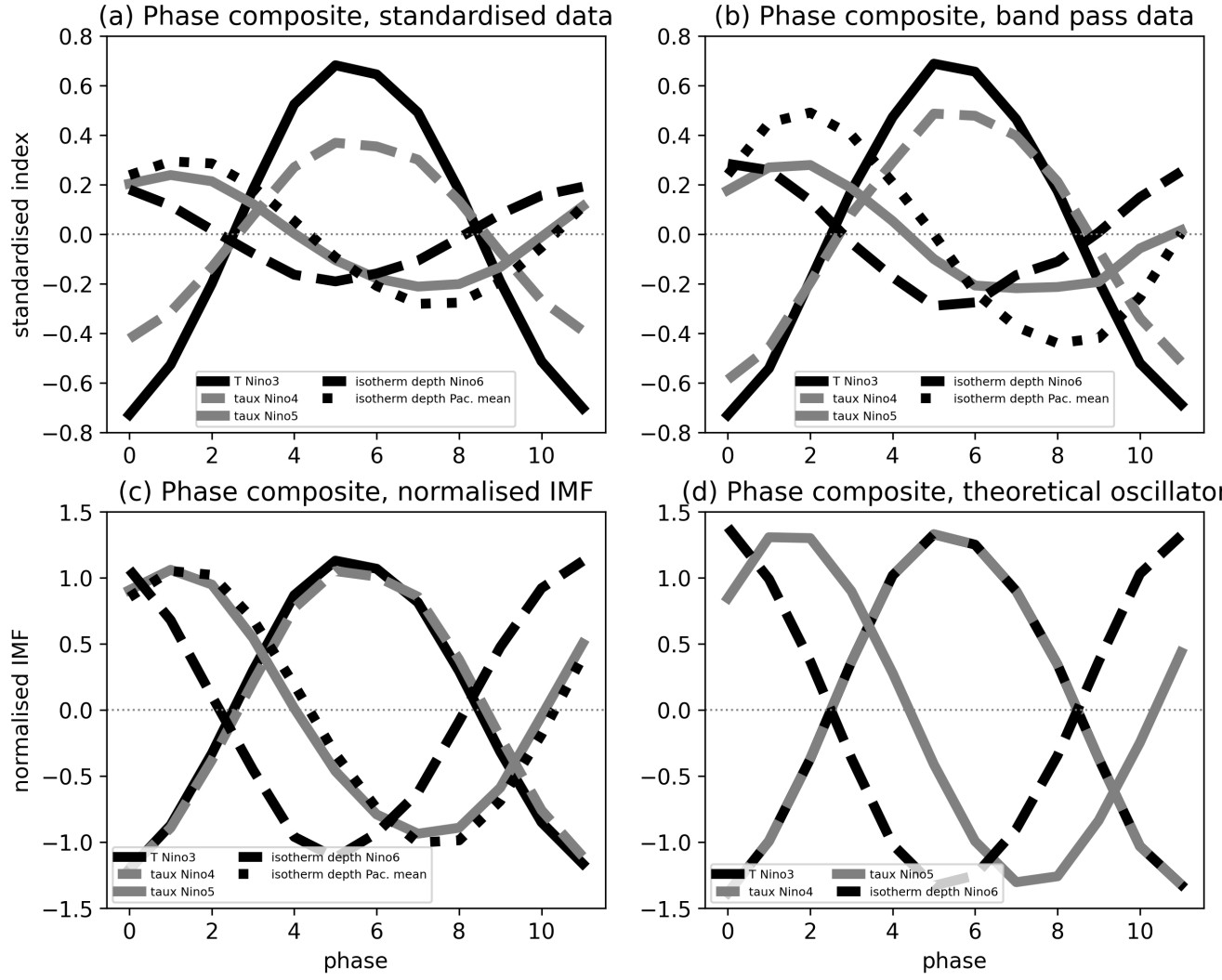

**Figure 3.** Phase composites of SST (Niño3) (black solid line), thermocline depth (Niño6) (black dashed line), thermocline depth (across tropical Pacific) (black dotted line), $\tau_x$ (Niño4) (grey dashed line), $\tau_x$ (Niño5) (grey solid line). All fields are composited over the phases of SST (Niño3), such that they fit the phases in Fig. 2. (a) composites of IMF13 for data divided by the standard deviation of corresponding timeseries (e.g., IMF13 (thermocline)/$\sigma$ (thermocline)); (b) composites of band-pass filtered (26.5-51 months) standardised timeseries; (c) as in (a) but IMF-timeseries are divided by IMF's standard deviation (e.g., IMF13 (thermocline)/$\sigma$ (IMF13 of thermocline)); (d) composites of the fields from the conceptual oscillator timeseries (Eqs. (5)-(8)), divided by standard deviations of corresponding timeseries (similar to (c)).

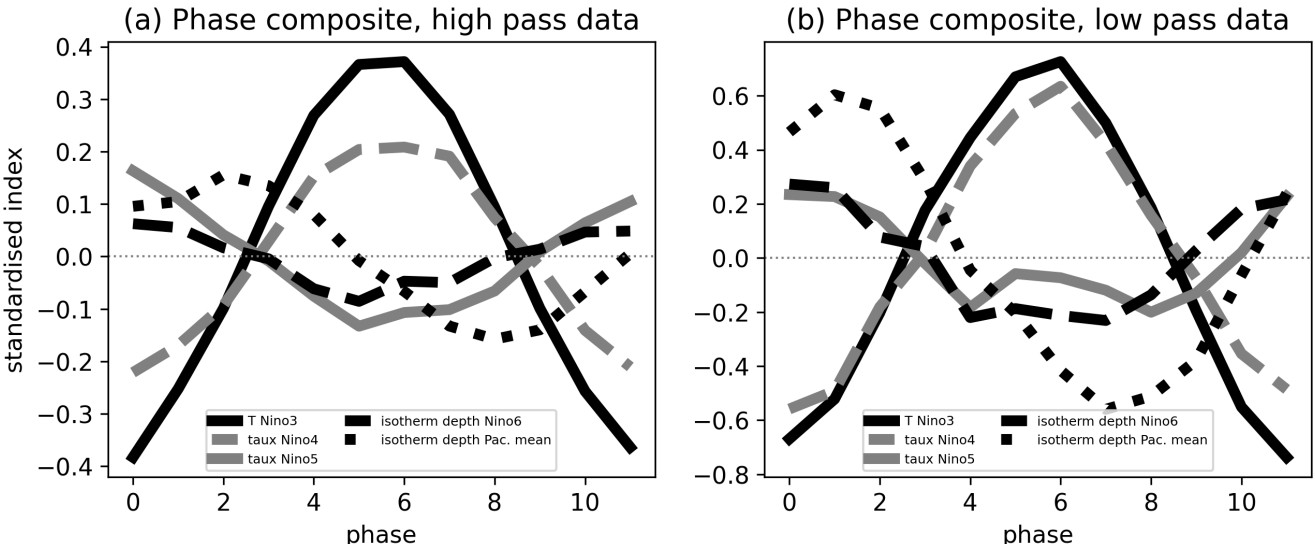

**Figure 4.** As in Fig. 3b, but for (a) high-pass filtered (11-19 months) data and (b) low-pass filtered (48-103 months) data.

## 3.2 Conceptual oscillator models of ENSO

**The Unified Oscillator**

As mentioned in the introduction, there are several different conceptual oscillator models (e.g., Wang, 2018): (i) the delayed; (ii) the recharge-discharge; (iii) the advective-reflective; (iv) the Western Pacific; and (v) the unified oscillator. The latter encompasses all the previous oscillators, and can be described by four relatively simple equations (Wang, 2001a)

$$\frac{\mathrm{d}T}{\mathrm{d}t} = a\tau_1 - b_1\tau_1(t-\eta) + b_2\tau_2(t-\delta) - b_3\tau_1(t-\mu) - \varepsilon T^3 \tag{1}$$

$$\frac{\mathrm{d}h}{\mathrm{d}t} = -c\tau_1(t-\lambda) - R_h h \tag{2}$$

$$\frac{\mathrm{d}\tau_1}{\mathrm{d}t} = dT - R_{\tau_1}\tau_1 \tag{3}$$

$$\frac{\mathrm{d}\tau_2}{\mathrm{d}t} = eh - R_{\tau_2}\tau_2 \tag{4}$$

where $T$ is SST in Niño3 region, $h$ is thermocline depth in Niño6 region, $\tau_1$ is $\tau_x$ in Niño4 region, $\tau_2$ is $\tau_x$ in Niño5 region, the constants are: $a = 1.5 \times 10^{-2}$ K m$^2$ N$^{-1}$ yr$^{-1}$, $b_1 = b_3 = 2.5 \times 10^2$ K m$^2$ N$^{-1}$ yr$^{-1}$, $b_2 = 7.5 \times 10^2$ K m$^2$ N$^{-1}$ yr$^{-1}$, $c = 1.5 \times 10^3$ m$^3$ N$^{-1}$ yr$^{-1}$, $d = 3.6 \times 10^{-2}$ N m$^{-2}$ K$^{-1}$ yr$^{-1}$, $e = 3 \times 10^{-3}$ N m$^{-3}$ yr$^{-1}$, the damping coefficients are: $\varepsilon = 1.2$ K$^{-2}$ yr$^{-1}$, $R_h = 5$ yr$^{-1}$, $R_{\tau_1} = R_{\tau_2} = 2$ yr$^{-1}$, and the delay times (lags) are: $\eta = 150$ days, $\delta = 30$ days, $\lambda = 180$ days, $\mu = 90$ days. The unified oscillator and all its special cases listed above have a timescale range between 2 and 5 years with the provided parameters.





Eqs. (1)-(4) represent different processes in the tropical Pacific (Wang, 2001a). Eq. (1) represents changes to the SSTs in the Niño3 region via: (i) positive Bjerknes feedback (first term on the right-hand-side (RHS)); (ii) negative feedback due to Kelvin wave reflection at the western ocean boundary (second term on the RHS); (iii) negative feedback due to wind-forced Kelvin wave contribution in the equatorial western Pacific (third term on the RHS); (iv) negative feedback due to Rossby wave reflection at the eastern ocean boundary (fourth term on the RHS); and (v) cubic damping term that limits anomaly growth (last term on the RHS). Eq. (2) represents changes to the off-equatorial thermocline depth (Niño6 region) via the wind stress in the equatorial central Pacific (Niño4 region) (first term on the RHS) and linear damping (second term on the RHS). Eq. (3) represents changes to the zonal wind stress ($\tau_x$) in the equatorial central Pacific (Niño4 region) via SSTs in equatorial eastern Pacific (Niño3 region) (first term on the RHS) and linear damping (second term on the RHS). Finally, Eq. (4) represents changes to the zonal wind stress in the equatorial western Pacific (Niño5 region) via off-equatorial western Pacific thermocline depth (Niño6 region; first term on the RHS) and linear damping (second term on the RHS).

However, the results from section 3.1 suggest that the average evolution on 2-3 year (average) timescale is different from the unified model and many other oscillator models discussed in Wang (2001a). In section 3.1 we have established that $\tau_x$ (Niño4) closely follows SST (Niño3) (Fig. 3a,b,c), i.e., they co-vary. Similarly, Fig. 3a,b,c suggests that thermocline depth (Niño6) is 'anticorrelated' with SST (Niño3) and $\tau_x$ (Niño4). Therefore, we now assume: (i) $\tau_1 \propto T$ by setting $\mathrm{d}\tau_1/\mathrm{d}t = 0$, yielding $\tau_1 = dT/R_{\tau_1}$, i.e., we replace $\tau_1$ by $dT/R_{\tau_1}$ everywhere; (ii) $h \propto -\tau_1, -T$ by setting $\mathrm{d}h/\mathrm{d}t = 0$, yielding $h = -c\tau_1/R_h = -cdT/R_hR_{\tau_1}$, i.e., we replace $h$ by $-cdT/R_hR_{\tau_1}$ everywhere; (iii) assume that the wave reflections at both eastern and western ocean boundaries are not necessary for this oscillator (typically assumed for the Western Pacific oscillator) by setting $b_1 = b_3 = 0$; (iv) for simplicity set $\lambda = \eta = \mu = 0$, and keep the other parameters the same. This yields a modified unified oscillator

$$\frac{\mathrm{d}T}{\mathrm{d}t} = a\frac{d}{R_{\tau_1}}T + b_2\tau_2(t-\delta) - \varepsilon T^3 \tag{5}$$

$$h = -\frac{cd}{R_hR_{\tau_1}}T \tag{6}$$

$$\tau_1 = \frac{d}{R_{\tau_1}}T \tag{7}$$

$$\frac{\mathrm{d}\tau_2}{\mathrm{d}t} = -e\frac{cd}{R_hR_{\tau_1}}T - R_{\tau_2}\tau_2. \tag{8}$$

The timescale of this oscillator (using the same parameters as above) is ~3 years, i.e., similar to IMF13 timescale. The parameters in Eqs. (5)-(8) can theoretically be adjusted (e.g., to change timescale) until the oscillatory behaviour breaks (or the 'system' becomes unphysical), which is true also for the unified oscillator in Eqs. (1)-(4). Note that if $\delta = 0$ this oscillator and its timescale remain similar (not shown), which is consistent with Weisberg and Wang (1997), who noted that Western Pacific oscillator remains qualitatively similar when all lags are set to 0. We will refer to Eqs. (5)-(8) as a *simplified West-Pacific (SWP) Oscillator* from hereon, since this model is somewhat similar to the Western Pacific oscillator developed by Weisberg and Wang (1997), except for the conditions $\tau_1 = dT/R_{\tau_1}$ and $h = -cdT/R_hR_{\tau_1}$. We view the SWP oscillator model as another special case of the unified oscillator.




As mentioned above, the Western Pacific oscillator can be captured by Eqs. (1)-(4) by setting $b_1 = b_3 = 0$, typically describing the following series of events (e.g., Wang, 2001b, their Fig. 2). During the warm phase of ENSO there is (*atmospheric*) condensation heating in the equatorial central Pacific (Niño4) (e.g., Deser and Wallace, 1990) that induces a pair of low pressure anomalies in the off-equatorial central Pacific region. This drives the westerly wind anomalies in the central Pacific (Gill 1980; see also yellow arrows in phases 3-6 in Fig. 2). The wind stress (Niño4) then leads to deeper thermocline and warmer SSTs in Niño3 region (solid contours and red colours in phases 3-6 in Fig. 2), i.e., a positive (Bjerknes) feedback further amplifies the signal (reflected in Eqs. (1) [first term], (3)).

However, these off-equatorial low pressure anomalies act to raise the off-equatorial thermocline via Ekman pumping (evident through dashed contours in phases 3-6 in Fig. 2) (e.g., Wang, 2001b). This brings colder waters to the off-equatorial ocean surface, cooling the SSTs in Niño6 region (captured in Eq. (2) via thermocline impacts on SSTs), and introduces a pair of off-equatorial high pressure anomalies in the western Pacific. These then induce easterlies in equatorial western Pacific (Niño5) (yellow arrows in phase 9 in Fig. 2; also reflected in Eq. (4)), which can ultimately cause upwelling of the cold waters. This upwelling then extends further eastward (dashed contours and blue colours in phases 8-11, 0 in Fig. 2; also reflected in Eq. (1) - third term on the RHS) with Kelvin wave propagation (reflected in delay time $\delta$), leading to a negative phase of ENSO, La Niña, and the cycle repeats.

What is different in the SWP oscillator compared with, e.g., the West-Pacific oscillator? The relationship between $h, T$ and $\tau_1$ in Eqs. (6), (7) suggests that on 16-51 months timescale the atmosphere over the central-eastern Pacific is co-varying (is in a 'steady state') with the underlying ocean in SWP oscillator (c.f., Wang, 2001a), and that eastern (co-varying with SSTs (Niño3)) and western thermocline depth anomalies also co-vary, but are anti-correlated, unlike in the West-Pacific oscillator. However, the western Pacific wind stress (Niño5) keeps an out-of-phase relationship with the SSTs (Niño3), suggesting an important role of the western Pacific in ENSO variability on 16-51 months timescale. Note that this analysis does not necessarily imply causality, and that other processes are likely needed to ultimately cause an ENSO event (e.g., related to a delayed oscillator or recharge-discharge oscillator discussed below), since the winds in the western Pacific are weak compared to the central Pacific winds. However, these winds are important for forcing eastward-propagating Kelvin waves, which can represent additional feedback for ENSO growth/decay (e.g., Wang, 2018). The western Pacific wind anomalies could also reflect 'state-dependent westerly wind bursts' in the western Pacific (Lopez et al., 2013; Lopez and Kirtman, 2014; Timmermann et al., 2018) that are important for the onset of ENSO events.

Eqs. (5)-(8) describe an average evolution of the parameters $(\tau_1, \tau_2, T, h)$, but not all individual events have this exact behaviour – partly reflected in the reduced amplitudes of $\tau_x$ and thermocline in Fig. 3a compared with the SST amplitude. The full timeseries of these parameters (Fig. 5) show that the relationship from Fig. 3a,b,c occurs often in the analysed period, especially for strong events. However, for weak events (middle panel in Fig. 5) the relationship is harder to establish – every event seems to be slightly different. This is somewhat consistent with Crespo et al. (2022), who noted that the dynamics of ENSO was different prior to 1970 relative to after 1970 with a dominant recharge-discharge oscillator in the latter period. Note that below we show that there is likely a relationship between the SWP and the recharge-discharge oscillators.





Fig. 3d shows a phase composite over the synthetic timeseries generated by solving equations (5)-(8) numerically with a first order Euler method (other more complex methods yield the same results). The phase composite (lines are the same as in Fig. 3c, except for exclusion of the black dotted line) clearly demonstrates that the lags between $\tau_x$ (Niño5), thermocline (Niño6), and SST (Niño3) or $\tau_x$ (Niño4) are consistent with observations (Fig. 3c).

As seen in section 3.1 (Figs. 3, 4), the dynamics on timescales of about 2-3 years (with a range of 1.5-4.5 years) is different from the higher and lower frequencies. Thus, the variables that are important for the SWP Oscillator are not necessarily important for the variability on shorter/longer timescales. On shorter/longer timescales, one can make the assumption $\tau_2 \propto h$, since $\tau_x$ (Niño5) closely follows thermocline (Niño6). Additionally, Fig. 4 shows that $\tau_1, T \propto -\tau_2, -h$, i.e., these quantities are anticorrelated. While this suggests that a unified model on low/high frequency timescales may be simplified to a delayed

oscillator (involving $T$ alone), we show below that also a recharge-discharge oscillator of Burgers et al. (2005) is present on those timescales.

Some of the results here are consistent with Graham et al. (2015), who considered 12-month low-pass filtered data to assess the conceptual unified model of Wang (2001a). They provided several suggestions to improve the unified model of Wang (2001a), e.g., to remove the tendency terms for both wind stress terms (in Eqs. (3), (4)). As mentioned above, Fig. 4 suggests

that such approximation would likely yield better results than the original unified oscillator model. However, this would only work in the high- and low-frequency ranges discussed here (Fig. 4), but not on timescales in between, i.e., timescales of about 1.5-4.5 years (Fig. 3). On those intermediate timescales, the SWP Oscillator presented here should be used instead. This means that on these intermediate timescales the tendency of the wind stress in the Niño5 region should be kept, but the one in the Niño4 region may be omitted.

Graham et al. (2015) also provided other suggestions, like adding a stochastic forcing term to wind stress equations (not attempted here for simplicity), and they also suggested using the thermocline depth in the Western Pacific averaged over a region that lies on the equator, rather than off-equator. From Fig. 2 we can see that this would likely yield similar results as thermocline depth averaged over the Niño6 region (to $\pm 1$ phase). They also suggested that a delayed oscillator is generally sufficient for describing ENSO variability, which has been also suggested above, but only for low/high frequency timescales.

**The Recharge-Discharge Oscillator**

The SWP Oscillator is based on Wang (2001a), which uses West-Pacific wind stress and off-equatorial thermocline depth as part of the overall ENSO oscillator. However, Burgers et al. (2005) suggested using only the Pacific mean thermocline depth and SSTs (Niño3), which led to the (simplest) recharge-discharge oscillator

$$\frac{\mathrm{d}}{\mathrm{d}t} \begin{pmatrix} T \\ h_{\mathrm{Pac}} \end{pmatrix} = \begin{pmatrix} -2\gamma & \omega_0 \\ -\omega_0 & 0 \end{pmatrix} \begin{pmatrix} T \\ h_{\mathrm{Pac}} \end{pmatrix} \tag{9}$$

where $T$ is SST (Niño3), $h_{\mathrm{Pac}}$ is the thermocline depth averaged across the tropical Pacific, $\gamma^{-1} = 24^{+22}_{-11}$ months is the damping timescale, and the period of this oscillator is $2\pi\omega^{-1} = 37^{+8}_{-4}$ months ($\approx 2\pi\omega_0$), with $\omega^2 = \omega_0^2 - \gamma^2$. The sub- and super-scripts in the period and damping timescale provide their 95% confidence levels. Note that the Burgers et al. (2005) model is based on

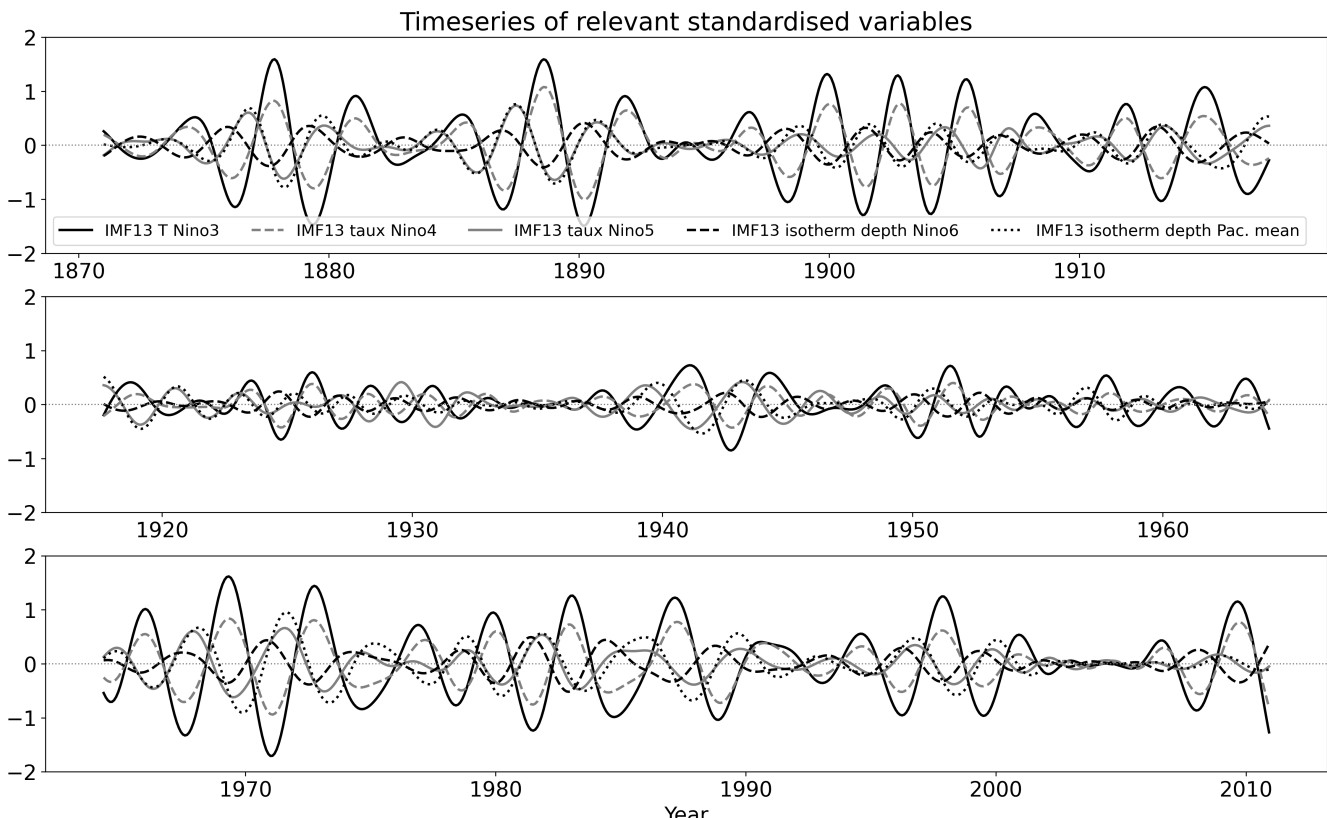

**Figure 5.** IMF13's standardised timeseries of SST (Niño3) (black solid line), thermocline depth (Niño6) (black dashed line), thermocline depth (across tropical Pacific) (black dotted line), $\tau_x$ (Niño4) (grey dashed line), $\tau_x$ (Niño5) (grey solid line).

the model developed by Jin (1997a, b). The recharge-discharge oscillator (e.g., Wang, 2001b) is related to Sverdrup transport (Sverdrup, 1947), which is associated with both zonal wind stress in the central Pacific (i.e., off-equatorial wind-stress curl)

and SSTs in the eastern Pacific. This is driving the recharge-discharge of the equatorial heat content that ultimately gives rise to oscillations.

Fig. 3a,b,c clearly demonstrates that SST (Niño3; black solid line) and thermocline depth averaged across the tropical Pacific (black dotted line) are $\sim 90°$ out of phase, consistent with the recharge-discharge oscillator of Jin (1997a, b); Burgers et al. (2005). We can also see (Fig. 3a,b,c) that $\tau_x$ (Niño5) (grey solid line) shows variability consistent with (proportional

to) the thermocline depth averaged across the tropical Pacific (black dotted line), i.e., one may assume $\tau_2 \propto h_{\text{Pac}}$ (in fact, their standardised values are essentially equal for IMF13). As with the SWP oscillator, the relationship between the relevant variables in the recharge-discharge oscillator on the 2-3 year timescale largely remains the same over time (Figs. 5, 1).

Wang (2001a) suggests that a recharge-discharge oscillator can be derived from the unified model by assuming $\tau_2 \propto h$. However, Fig. 3 suggests that on 2-3 year timescale the Pacific mean thermocline ($h_{\text{Pac}}$) and wind stress in the western Pacific



($\tau_2$) co-vary, thus the approximation should be $\tau_2 \propto h_{\mathrm{Pac}}$ to retrieve the Burgers et al. (2005) model. The recharge-discharge and
SWP oscillators are thus likely related on 2-3-year timescale, which could suggest a role of western Pacific wind-forced Kelvin
waves in recharge-discharge process (and vice versa). Thus, another way of looking at the two oscillators to combine them, e.g.:
(i) by using the recharge-discharge oscillator (Eq. 9) or SWP oscillator (Eqs. 5-8) and setting $\tau_2 = \alpha_1 h_{\mathrm{Pac}}$ or $h_{\mathrm{Pac}} = \alpha_2 \tau_2$ (with
$\alpha_{1,2}$ a constant); (ii) by replacing $h$ in the unified model with $h_{\mathrm{Pac}}$ and re-tuning the parameters of the model; or (iii) setting
$\tau$ during the recharge/discharge process to a non-zero value $\tau_2$ (see, e.g., Fig. 1b,d in Jin 1997a), i.e., potentially re-tuning the
recharge-discharge oscillator. It may also be necessary to reassess the relative importance of different variables in the oscillator
models.

The relationship between the SSTs (Niño3) and the Pacific mean thermocline depth remains the same at lower and higher
frequencies (see black solid and dotted lines in Fig. 4), suggesting that the recharge-discharge oscillator (Jin, 1997a, b; Burgers
et al., 2005) is present on all timescales considered here. This makes the relationship between the SWP and recharge-discharge
oscillators less clear on these longer/shorter timescales (Fig. 4). This suggests that the relationship between the two models
is non-trivial, and can depend on the timescale (Figs. 3, 4). However, this also suggests that the unified oscillator captures
different dynamics on different timescales, and that this is likely related to different behaviour in the western Pacific on different
timescales. These relationships should be explored further in the future.

Overall, this section has demonstrated that MEMD can extract physical modes of variability from the data. Namely, we have
identified a SWP Oscillator in the frequency range 1.5-4.5 years in observations/reanalyses (Fig. 3c) and through a conceptual
oscillator (equations (5)-(8); Fig. 3d) by modifying the unified oscillator conceptual model (equations (1)-(4); Wang 2001a). At
lower and higher frequencies we have found different behaviour, likely related to different dynamical processes in the western
Pacific on those timescales. Note, however, that a quasi-periodic behaviour in observations on low/high frequency timescales
has not been detected. We have also found a recharge-discharge oscillator (Jin, 1997a; Burgers et al., 2005) on *all* timescales,
suggesting similar dynamical processes in the eastern Pacific on all timescales considered, and a complex relationship with the
unified model (Wang, 2001a). This suggests that a reassessment of the oscillator models and their links across timescales may
be necessary (left for future work).

## 4 Further implications of understanding ENSO's intrinsic variability

The previous section established that MEMD provides physical modes of variability and a red noise test can be used on
MEMD's IMFs to identify oscillatory behaviour (when data has typically red noise distribution). Here we discuss the repre-
sentation of the SWP and recharge-discharge oscillators in a climate model (section 4.1), as well as ENSO predictability using
a simple statistical model (section 4.2), both based on the knowledge gained in the sections above.

Of course, the implications of MEMD (in general) and better understanding of ENSO variability are not limited to the cases
presented here and could be explored in many other applications in the future.

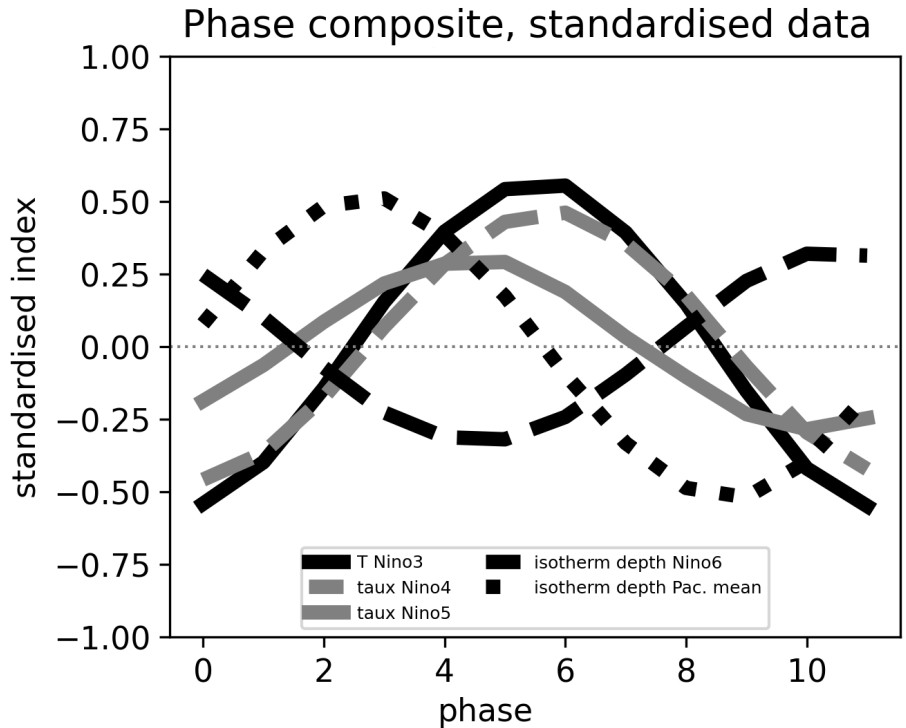

**Figure 6.** As in Fig. 3a, but for the corresponding IMF15 fields in the NorCPM1 (ensemble member 1).

## 4.1 Model evaluation

Since the extracted modes are physical and the significant modes quasi-periodic, we can now repeat the analysis in a climate model. Here we use the NorCPM1 historical simulation (first ensemble member; the other ensemble members show qualitatively similar results) to test the ENSO dynamics in different frequency bands (similar to Fig. 3). The MEMD analysis yields

two significant (quasi-periodic) modes also in the model: IMF14 with a timescale of 28 months (range: 24.5 - 32.5 months), and IMF15 with a timescale of 39 months (range: 33 - 48 months). As in the observations the two modes show similar behaviour and fall well within the 16-51 month range of observed quasi-periodic behaviour.

     Fig. 6 shows the model's IMF15 phase composites, a figure equivalent to observations' Fig. 3a (note that as in observations, also here the same band pass filter yields similar results to an IMF). While the recharge-discharge oscillator (e.g., Burgers

et al., 2005) is well represented in the model (black solid line for SST (Niño3) and black dotted line for the tropical Pacific mean thermocline depth), the SWP oscillator (other lines and black solid line) shows slightly different behaviour compared with observations. This is especially evident in the $\tau_x$ (Niño5 region; grey solid line), which peaks just before the SST (Niño3), leading it by $\sim$1 phase (Fig. 6), whereas in observations (Fig. 3a) it peaks well before the SST (Niño3), leading it by $\sim$3 phases. At the same time, $\tau_x$ (Niño5; grey solid line) is anticorrelated with thermocline depth (Niño6; black dashed line),

which is different from observations where thermocline depth (Niño6) was anticorrelated with SST (Niño3) and $\tau_x$ (Niño4).



This suggests an issue in the model's representation of the ENSO dynamics on 2-3 year timescale, especially in the Western Pacific, since the recharge-discharge mechanism, which is stronger in the eastern Pacific, is correctly represented. The $\tau_x$ (Niño4) shows similar evolution as in the observations, but its strength is stronger in the model (compare grey dashed lines in Fig. 3a and Fig. 6), suggesting a stronger feedback to the SST (Niño3) or less varied dynamical processes. Consistently, there is also a stronger relation of the SST (Niño3) to the tropical Pacific mean thermocline depth (larger amplitude relative to the SSTs compared with observations; compare black dotted lines in Figs. 3a, 6), likely suggesting a stronger (more dominant) recharge-discharge oscillator in the model.

The strong East-Pacific thermocline-SST feedback in the model was also identified as a general model bias in Chen et al. (2021), who also identified model biases in the western Pacific (e.g., SST anomalies during ENSO events extending too far west). These are likely responsible for different model behaviour in the western Pacific compared with observations. We also found differences at low and high frequencies between the model and the observations (in the unified oscillator), further suggesting potential issues with the dynamics of ENSO on different timescales in the model.

This analysis suggests that MEMD (and similar methods) could be used also to evaluate models. Here, it confirms that the analysed model has correct periodic behaviour (perhaps too periodic), however it may be struggling with the exact physics that lead to the onset of ENSO (or ENSO diversity), which should be considered for future model improvements (see also Guilyardi et al. 2020; Planton et al. 2021). Nonetheless, since (i) the recharge-discharge process is well represented in the model; (ii) the model exhibits quasi-periodic behaviour; and (iii) SST-variability alone could be sufficient for describing/predicting ENSO (see section 4.2; Graham et al. 2015) – the NorCPM1 (prediction system) should be able to predict ENSO, especially on timescales ~2-4 years (left for future work).

## 4.2 Prediction

The knowledge gained from the MEMD analysis and its significance can also be used for improving predictions or finding time-windows in which prediction works well. To do this, we take the three (input) timeseries: SST (Niño3), $\tau_x$ (Niño5), and thermocline depth (Niño6) to predict the SST (Niño3). We included thermocline depth (Niño6) as a predictor as this improved the results compared with using just SST (Niño3) and $\tau_x$ (Niño5) as predictors, which are the two main quantities in the SWP oscillator (Eqs. (5) - (8)). On the other hand, we do not use $\tau_x$ (Niño4) as it is clear from Figs. 3, 4 that it is somewhat redundant (has the same evolution as the SST (Niño3)), and including it actually reduces predictability (not shown). Similarly, predictions from tropical Pacific mean thermocline depth are not considered here as they did not improve predictions (not shown). We test the prediction for raw data and for data in the 26.5-51 months band, where the SST (Niño3) and PC1 of tropical Pacific SSTs were both significant (i.e., IMF13 timescale band). An extension to 16-51 months band (used above) reduces the skill significantly and is hence not pursued further (not shown).

We first prepare timeseries that we can input into the statistical prediction model(similar to Omrani et al. 2022) based on multi-linear regression the following way. First we lag the input timeseries relative to the SST (Niño3) based on the lag of the maximum correlation of data band-pass-filtered in 26.5-51 months band, capturing the first maximum and minimum of the correlation between the timeseries, i.e., including a full wave/oscillation. This generates 6 predictors for the SST (Niño3)



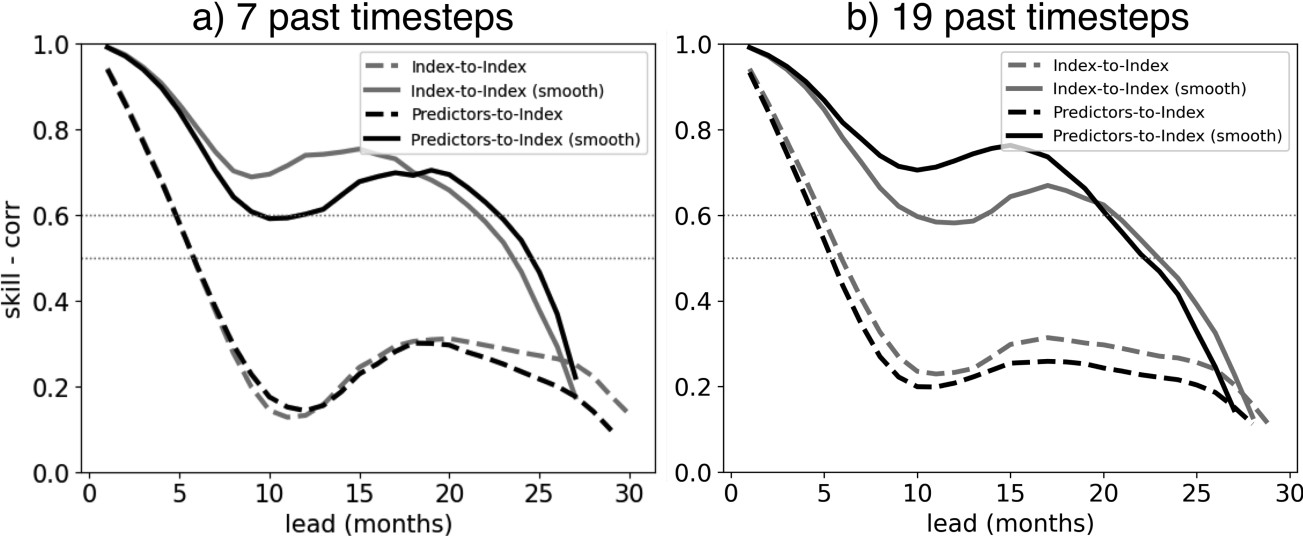

**Figure 7.** Correlation skill score of a multi-linear regression model for predicting SST (Niño3) from (a) 7 past timesteps (in months), and (b) 19 past timesteps (in months). The dashed lines are for raw predictions (without smoothing), and solid lines are for a (smoothed) band-pass filtered prediction (26.5-51 months). Note that we smooth the raw prediction to get the smooth-prediction skill score. The grey lines are for a prediction from SST (Niño3) alone, whereas black lines are for prediction from SST (Niño3), $\tau_x$ (Niño5) and thermocline depth (Niño6). For further details see the text.

(the predictand): SST (Niño3) max at lag 0 months, SST (Niño3) min at lag -14 months, $\tau_x$ (Niño5) max at lag -9 months, $\tau_x$ (Niño5) min at lag -34 months, thermocline depth (Niño6) min at lag -13 months, and thermocline depth (Niño6) max at lag -37 months. Note that this yields better results than if only half of the lags (i.e., half of the oscillation) were used (not shown). The results from the prediction based on these 6 predictors are compared with the prediction of SST (Niño3) from SST (Niño3) at lag 0 alone.

By lagging the data we generate timeseries of 6 predictors and the predictand that have maximum correlations at lag zero. We can then input the timeseries into the multi-linear regression model and predict SST (Niño3) at different lead times (1, 2, 3, ... , 30 months). For each lead time we construct a new prediction model. To improve predictions we use the predictors' data from several past timesteps (i.e., use each of the predictors at months ..., -19, -18, ..., -1 relative to the predictand – this can increase the 'number' of predictors significantly) and we test sensitivity to the number of the past timesteps used. The

prediction model is repeated for all months within the 1871-2010 period (except for the timesteps used for lagging the data). We use 60% of data for training and 40% of the data for testing.

   The skill score is assessed using the correlation between the true SST (Niño3) and its prediction in the testing dataset. The correlation is performed on raw data and on band-passed (26.5-51 months) data. Note that the prediction model is always used on raw data, and the prediction itself is later smoothed by using the band-pass filter.





Fig. 7 shows SST (Niño3) prediction skill for 30 lead months using (a) 7 months of (past) data as predictors, and (b) 19 months of (past) data as predictors for predictions from SST (Niño3) alone (grey lines), and prediction from all six predictors (black lines). Dashed lines represent the raw prediction skill, and solid lines represent prediction skill of smoothed data (26.5-51 months band pass). This reveals that there is extended prediction skill in the smoothed data to ∼22 months lead time (skill score over 0.6) compared with raw prediction skill (∼5 months) in the regression model. This is consistent with the oscillatory

behaviour in that timescale-band (identified in Sect. 3).

    Fig. 7a reveals that we can get a good prediction of SST (Niño3) from SST (Niño3) alone by using 7 months of (past) data (Fig. 7a), which can be slightly extended by a month or two when all predictors are considered (black solid line crosses the 0.6-skill-score line at a slightly longer lead time beyond 20 months). By using 19 months of (past) data (Fig. 7b), we can improve raw predictions (dashed lines) for lead times 10-15 months, and the skill-score of smoothed predictions using all

550 predictors (black solid line) improves relative to Fig. 7a. The latter also yields slightly better skill-score than prediction of SST (Niño3) from SST (Niño3) in Fig. 7a (grey solid line), i.e., the same skill score can last ∼1-2 months lead-time longer in Fig. 7b (black solid line) – until lead time ∼20 months. This suggests that statistical predictions of ENSO can be well captured by SST (Niño3) alone, which is perhaps consistent with Graham et al. (2015), who suggested that a delayed oscillator is sufficient for describing ENSO. Note that root-mean-square error was not sensitive to the choice of predictors, hence it is not shown.

While this work suggests that there is indeed potential for very good predictability (for over 20 months lead time) of ENSO on timescales 26.5-51 months, such predictability can only provide the sign of ENSO in this frequency band, and it can suggest a higher chance of positive or negative ENSO event, but it cannot provide the actual magnitude and prediction of the full ENSO event. A lot of work has been done on dynamical and statistical model predictions of ENSO, which can predict an ENSO event reasonably well up to 6 months ahead (sometimes more) from raw data (e.g., L'Heureux et al., 2020; Dijkstra et al., 2019).

However, we show that such models might have a predictable (up to 2 years) component on timescales 25.6-51 months. This is also because models can simulate ENSO variability on this timescale well (as established in section 4.1). Perhaps ENSO events that can be predicted up to 2 years ahead (e.g., Chen et al., 2004; Park et al., 2018) have a strong component of this ∼2-4.5 year mode of variability.

    Note that here we computed 'prediction' skill without any preconditioning or a selection of a season, therefore better skill

scores may be obtained if the model is initialised in relevant seasons or based on certain precursors (left for future work).

## 5   Conclusions

In this study we have used observational and reanalysis products to study the variability in the tropical Pacific on interannual timescales (i.e., ENSO). To do this, we have used a recently developed method for identifying intrinsic variability of multivariate systems, the multivariate empirical mode decomposition (MEMD; Rehman and Mandic 2010). The method can objectively

identify variability on different timescales, i.e., it works as a band-pass filter. We can then use a red noise significance test to extract quasi-periodic modes of variability in our data.



Here, we identified a quasi-periodic behaviour on timescales of about 3 years (26.5-51 months; arguably also 16-51 months as discussed in previous sections). This timescale falls within the typical timescale range of ENSO, i.e., 2-8 years. While the ENSO quasi-periodic variability is a well-known feature, an identification of a range of timescales of the quasi-periodic modes (e.g., 26.5-51 months) has led to a few interesting results.

By analysing composites of the thermocline depth, surface wind stress and the sea surface temperature (SST), we have shown that the ∼3-year ENSO variability generally follows an oscillator model that is a subset of the unified oscillator (Wang, 2001a). We refer to this oscillator as a simplified West-Pacific (SWP) Oscillator, as it is slightly different to the other established oscillator models (i.e., the Western Pacific, delayed, advective-reflective, recharge-discharge oscillators). We also find the recharge-discharge oscillator of Burgers et al. (2005), which is described by a slightly different set of equations than the unified oscillator (i.e., uses Pacific mean thermocline depth instead of average over the Niño6 region). However, the recharge-discharge oscillator is present also on other timescales (not just on ∼3-year timescale), unlike the SWP Oscillator. This suggests similar behaviour in the eastern Pacific on all timescales considered, but different behaviour in the western Pacific. This also suggests that the relative role and phasing of relevant variables in the tropical Pacific may change depending on the timescale, which may ultimately be important for interactions across scales that give rise to, e.g., an ENSO event, thus these relationships should be explored further in the future.

The SWP Oscillator (Eqs. (5)-(8)) resembles the Western Pacific Oscillator of Weisberg and Wang (1997), in that it keeps the evolution of wind stress in the western Pacific (Niño5). However, based on the results from the reanalyses/observations (Fig. 3a,c) we assume that wind stress in the Niño4 region and thermocline depth in Niño6 region can be modelled through the SSTs in the Niño3 region (i.e., their tendency can be omitted). This is because wind stress (Niño4) closely follows SSTs (Niño3) and thermocline depth (Niño6) is largely anticorrelated with SSTs (Niño3). This suggests co-variability of the atmosphere and ocean on interannual timescales in the central-eastern Pacific, but an out-of-phase relationship with the western Pacific atmosphere-ocean processes, which are important for the generation of Kelvin waves at the western boundary. The latter provides additional feedbacks (e.g., Wang, 2018) for the variability in the eastern Pacific, suggesting that the western Pacific processes are important for ENSO variability. There might be an additional link to the 'state-dependent westerly wind bursts' that are important for ENSO onset (e.g., Lopez et al. 2013; Lopez and Kirtman 2014; Timmermann et al. 2018). However, since the amplitude of the western Pacific wind stress anomalies is generally small it is likely that other processes are important as well – e.g., recharge-discharge of the heat content (see below).

The evolution of the SWP Oscillator on timescales of 16-51 months can be described in the following way (c.f., Wang 2001b; Figs. 2, 3a,c):

1. warm phase of ENSO is associated with westerly winds in Niño4, warmer SSTs and deeper thermocline in Niño3 (phases 4-6 in Fig. 2);

2. this is associated with off-equatorial pair of low pressure anomalies in central Pacific, which can cause upwelling (via Ekman pumping) of the off-equatorial thermocline (becoming shallower), extending further west (into the Niño6 region). This largely occurs together with the warmer SSTs in Niño3 (see evolution of dashed contours in phases 3-8 in Fig. 2);



3. the colder waters caused by upwelling of the thermocline in Niño6 lead to off-equatorial pair of high pressure anomalies in the western Pacific, which ultimately gives rise to easterlies in the western Pacific (Niño5) (e.g., yellow arrows in phase 9 in Fig. 2);

4. these easterlies can cause upwelling of cold waters (shallower thermocline) in the equatorial western Pacific, which can then propagate as Kelvin waves further east (dashed contours in phases 6-10 in Fig. 2), bringing colder waters to the eastern Pacific and initiating the cold phase of ENSO (phases 10, 11, 0 in Fig. 2);

5. the cycle reverses and ultimately repeats.

The recharge-discharge oscillator of Burgers et al. (2005) is also present on the 16-51 months timescale throughout the analysed period (Fig. 5) and should be considered as part of (or alternative to) the SWP Oscillator (or vice versa). This is because the wind stress (Niño5) and the Pacific mean thermocline depth co-vary and are likely related to each other – their relationship should be explored further in the future. The evolution of the recharge-discharge oscillator can be summarised as follows (e.g., Burgers et al. 2005; Wang 2001b; Figs. 2, 3a,c): (i) during the warm phase of ENSO (phases 4-6 in Fig. 2) there is westerly wind stress in the central Pacific, warmer SSTs and deeper thermocline in the eastern Pacific; (ii) this is associated with divergent Sverdrup transport that ultimately drives discharge of the equatorial heat content, which leads to (climatological) upwelling of cold waters throughout the equatorial Pacific (phase 9 in Fig. 2); (iii) this initiates a cold phase of ENSO (phases 10,11,0 in Fig. 2), the cycle reverses and ultimately repeats.

Exploring the representation of ENSO's intrinsic variability in a climate model (NorCPM1) showed that the model has a similar quasi-periodic behaviour as the observations/reanalyses, with largely accurate (though too strong) recharge-discharge oscillator. However, SWP Oscillator is not really present, i.e., the dynamics is different, since wind stress (Niño5) peaks too late and thermocline depth (Niño6) peaks too early (Fig. 6). Similarly, the co-variability of the recharge-discharge and SWP oscillators is absent in the model. Therefore, we speculate that climate models (likely not just NorCPM1) may struggle with the relationship between the western and eastern Pacific atmosphere-ocean processes.

By constructing a statistical prediction model, we have shown that SST variability on timescales of 26.5-51 months in the tropical Pacific may be predictable for up to 22 months in advance (Fig. 7). Also, given that NorCPM1 can reproduce the recharge-discharge oscillator well and that it exhibits periodic variability on timescales 2-4 years, we believe that it should be able to predict the variability on the 2-4 year timescale reasonably well – this is likely true for other models as well (left for future work). However, such prediction can only tell us that there is a higher chance of a certain ENSO event and it cannot yield a prediction of full ENSO amplitude, i.e., this remains challenging. This is because an ENSO event ultimately depends on variability on all timescales (and their interactions), thus its peak magnitude and timing will likely differ from the one identified on a specific (band-passed) timescale.

Therefore, a better understanding of ENSO variability on different timescales is important for a better representation of ENSO dynamics in the climate models. Additionally, it is important to understand ENSO impacts both locally and remotely (teleconnections) on different timescales. The latter can be achieved through similar analysis as in this study, but including other fields and other (remote) regions in the analysis (e.g., MSLP in the North Atlantic). Also, future studies should involve



an examination of sensitivity and causal links (not established here) between different variables (and their links across scales)
within the Tropics and beyond (e.g., Runge et al., 2015; Jajcay et al., 2018; Jenney et al., 2019; Kretschmer et al., 2021), as
well as dedicated model-experiments.

Overall, this study has analysed the variability in the tropical Pacific, identified a quasi-periodic mode of variability (on
$\sim$3-year timescale) and related its physics to the SWP and recharge-discharge oscillators, which are likely related to each other
on this timescale. Variability on this timescale may be predictable far in advance, which calls for further investigations of the
tropical Pacific variability and related teleconnections, their prediction, and for further model improvements (see also Chen
et al., 2021; Lee et al., 2021).

*Code availability.* EMD and MEMD Python codes are available on Github (https://github.com/laszukdawid/PyEMD,
https://github.com/mariogrune/MEMD-Python-). Other scripts are available upon request.

*Data availability.* NOAA20CR data can be downloaded from https://psl.noaa.gov/data/gridded/data.20thC_ReanV3.html;
SODA2 data can be downloaded from http://apdrc.soest.hawaii.edu/dods/public_data/SODA/soda_pop2.2.4;
HadISST data can be downloaded from https://www.metoffice.gov.uk/hadobs/hadisst;
and NorCPM1 data is a part of the CMIP6 project (available on https://esgf-node.llnl.gov/search/cmip6/).

**Appendix A: MEMD for 3D fields**

To find the intrinsic variability of our 3D field, i.e., $\mathbf{A}'(t,y,x)$ mentioned in Sect. 2.2, we first reduce dimensionality of our
data by decomposing it using the singular value decomposition (SVD), which yields spatial patterns of our data (empirical
orthogonal functions, EOFs) and corresponding timeseries (principal components, PCs). First we multiply $\mathbf{A}'$ by $\sqrt{\cos\phi}$ (area
weighting), divide by standard deviation ($\sigma$), and reshape $\mathbf{A}'$ from $(t,y,x)$ to $(t,xy)$. Then $\mathbf{A}'$ can be expressed with a singular
value decomposition as

$$\sigma^{-1}(xy)\mathbf{A}'(t,xy) = \mathbf{U}\mathbf{\Sigma}\mathbf{V}^T \tag{A1}$$

where $\mathbf{U}$ and $\mathbf{V}$ represent left and right singular vectors, i.e., the normalised PCs and EOFs, $\mathbf{\Sigma} = \sqrt{(N-1)\mathbf{\Lambda}}$ is a diagonal
matrix with square roots of variance explained of each mode (denoted $\mathbf{\Lambda}$, i.e., eigenvalues) along the diagonal, $N$ is the number
of spatial points, and superscript $T$ denotes transpose. In order to keep the information of the variance explained of each mode
within the data we define PCs as $\mathbf{U}\mathbf{\Sigma}/\sqrt{N-1}$, such that $\sigma^{-1}(xy)\mathbf{A}'$ can be represented as a function of PCs and EOFs (recall
EOFs are in $\mathbf{V}$)

$$\sigma^{-1}(xy)\mathbf{A}'(t,xy) = \sqrt{N-1} \sum_{m=1}^{m=20} \text{EOF}(m,xy)\text{PC}(m,t) \tag{A2}$$





where $m$ corresponds to PC-number and is ordered according to the eigenvalues ($m = 1$ for the largest eigenvalue). We retain only the leading 20 PCs for the analysis (they generally describe the majority of the variance in $\mathbf{A}'$).

Now we can use the 20 PCs (PC(m,t)) as input to MEMD algorithm (for details on algorithm itself see Rehman and Mandic 2010). This algorithm finds common timescales (i.e., Intrinsic Mode Functions, IMFs) within the 20 PCs and splits each PC into several IMFs (the number of IMFs is not predetermined). Thus, each PC can be represented as a sum of IMFs

$$\text{PC}(m,t) = \sum_{s=1}^{s=s_{max}} \text{IMF}(s,m,t) \tag{A3}$$

where $s$ corresponds to IMF-number and is ordered according to the timescale ($s = 1$ for the shortest timescale, $s_{max}$ for the longest timescale, which is usually a trend or a residual). Eq. (A3) shows that each PC is a superposition of different IMFs and

with it also a superposition of modes of variability in the selected filed(s) with different timescales.

Each PC is associated with a spatial pattern (EOF($m,xy$)), which allows a reconstruction of the time-space $(t, xy)$ structure/evolution for each IMF, yielding IMFs of initial dataset $\mathbf{A}'$ (**IMFA**). We do this by multiplying PCs for each IMF with corresponding EOFs and summing over all 20 PCs/EOFs (similar to Eq. A2)

$$\sigma^{-1}(xy)\textbf{IMFA}(s,t,yx) = \sqrt{N-1} \sum_{m=1}^{m=20} \text{IMF}(s,m,t)\text{EOF}(m,xy). \tag{A4}$$

Here note that to get **IMFA** in the units of the input field we must multiply it by the field's standard deviation as the input data for the SVD algorithm was standardised (Eq. A1). From here we can reconstruct $\mathbf{A}'$ by summing over all **IMFA**

$$\sigma^{-1}(xy)\mathbf{A}'(t,xy) = \sum_{s=1}^{s=s_{max}} \textbf{IMFA}(s,t,yx) \tag{A5}$$

and ultimately one can also reshape $\mathbf{A}'$ from $(t, xy)$ to $(t, y, x)$. The importance of each **IMFA** for $\mathbf{A}'$ can be assessed by computing variance explained of each **IMFA** or other significance methods. To find **IMFA** modes (and grid-points) that correspond

to potentially oscillatory behaviour we must perform a red noise test (see Appendix B).

Note that from here on (and in the main text) **IMFA**s are referred to as IMFs for simplicity.

## Appendix B: Significance tests

Typically we can test if the modes (IMFs) are different from white or red noise, if we expect such distribution in our data. In the climate system, more often than not we expect the variables to behave as white or red noise. The IMFs that are significant

(i.e., different from both red and white noise) are likely representing oscillations instead, which can suggest higher potential for predictability of processes that correspond to the timescale of the significant IMF. Thus, this distinction is very important in climate system science. Therefore, we discuss the white and red noise tests (for 1D data, i.e., timeseries) below, whereas the robustness of IMFs and the significance tests are briefly mentioned where relevant.

Note that the white and red noise tests are performed on 1D timeseries, hence EMD (univariate decomposition) is used to

show their performance. The multivariate data (via MEMD) is later analysed with only one test - the simplest and most relevant one.





## B1 White noise test

The white noise significance test has been derived by Wu and Huang (2004), who showed that the energy density function of $\text{s}^{th}$ IMF ($E_s$) is

$$E_s = \frac{1}{L}\sum_{j=1}^{L}[C_s(j)]^2, \tag{B1}$$

where $C_s(j)$ is the $\text{s}^{th}$ IMF at time-step $j$ ($j = 1,...,L$), and $L$ is the length of the timeseries. Wu and Huang (2004) further showed that the total energy density of the $\text{s}^{th}$ IMF can then be expressed as

$$LE_s = \int S(\nu)_s d\nu \tag{B2}$$

where $\nu$ is frequency, and $S(\nu)_s$ is the power spectrum of the $\text{s}^{th}$ IMF. From this they showed that for white noise

$$\ln E_s \approx -\ln T_s \tag{B3}$$

where $T_s$ is the average period of the $\text{s}^{th}$ IMF.

Note that frequency (and thus also period) of each IMF is computed using Hilbert transform by first generating an analytical signal (e.g., Huang et al., 1998)

$$Z(t) = X(t) + \mathrm{i}Y(t) = |Z(t)|\mathrm{e}^{\mathrm{i}\theta(t)} \tag{B4}$$

where $X(t)$ is our IMF timeseries, $Y(t)$ is its Hilbert transform, $Z(t)$ is the analytical signal, and $\theta(t) = \arctan(Y(t)/X(t))$ is instantaneous phase. Instantaneous frequency can then be computed by taking a time-derivative of the phase, i.e., $\nu = \mathrm{d}\theta(t)/\mathrm{d}t/2\pi$, and the average frequency of each IMF is computed by averaging instantaneous frequency in time (note that period $= 1/\nu$).

The relationship between the logarithms of energy density and average period of the IMFs (Eq. B3) is then used in Fig. B1a (black solid line) to test whether an IMF (using EMD decomposition of Niño3 index) is different from white noise or not. The mode is significant with respect to white noise if it exceeds one-tailed 95th threshold (denoted by black dotted line). The percentile range serves as a significance test, i.e., if IMFs from our data are above, e.g., 95th percentile they are significant at 95th percentile level. The percentile range can be expressed analytically as (Wu and Huang, 2004)

$$\ln E_s = -\ln T_s \pm p\sqrt{\frac{2}{L}}\exp(\ln T_s/2) \tag{B5}$$

where $p$ denotes a threshold ($p = 1.645$ for one-tailed 95th percentile). Note that typically the number of degrees of freedom for white noise data is expected to be equal to the total energy density of the $\text{s}^{th}$ IMF (i.e., $LE_s$; Wu and Huang 2004).

Alternatively, we can test whether the input data is different from white noise by constructing multiple ($I$) realisations of synthetic white noise timeseries $w_i$ ($i^{th}$ random normally distributed timeseries with standard deviation $\sigma$ of 1). Then we can compute its IMFs via EMD (section 2.2) and we can repeat the process $I$-times. Employing Eq. B1 on these IMFs yields scattered grey dots in Fig. B1a, where their mean and 95th percentile are shown as grey solid and dotted lines, respectively.





A comparison with the IMFs from the input data (Niño3 index; blue dots in Fig. B1a) reveals that many IMFs lie outside the white noise range and that the overall data (blue dots) distribution does not resemble the white noise (grey dots) distribution (not noted in Wu and Huang 2004). This suggests that a white noise test for such data is not a good test. Indeed, atmosphere-ocean coupled systems, such as ENSO, can often be represented as a red noise process (e.g., Hasselmann 1976; Frankignoul

and Hasselmann 1977), thus we now turn to a similar test, but for data distributed as red noise.

## B2    Red noise test

### B2.1    Synthetic red noise data

To test if our data (e.g., Niño3 index) is purely red noise or it has inherent oscillations that we can identify through the EMD, we generate $I$-realisations of synthetic red noise timeseries $x$ (AR(1) process) as (e.g., Gilman et al. 1963)

$$x_{i,j+1} = rx_{i,j} + \sqrt{1-r^2}\, w_{i,j+1} \tag{B6}$$

where $r$ is lag-1 auto-correlation from our data (e.g., Niño3 index), $w$ is white noise (as in Appendix B1), $i$ runs over $I$ realisations of synthetic red noise data, and $j$ ($j > 1$ and $j \leq L$; with $L$ length of our data, e.g., the length of Niño3 record) is an index that runs over the time-steps (one time step is one time unit, e.g., 1 month). For $j = 1$ (the first time-step) we set $x_{i,1} = w_{i,1}$.

Once we obtain the red noise timeseries $x_i$ we can compute its IMFs via EMD (section 2.2) and we can repeat the process $I$-times (as for the white noise; Appendix B1). This yields the pink scattered dots in Fig. B1b. The mean over $I$ cases for each IMF (frequency band) is shown by pink solid line and the (one-tailed) 95th percentile across the $I$ cases are shown by pink dotted line.

Note that Franzke (2009) used a similar approach for indices such as the North Atlantic Oscillation, and found a simple

relationship between the power spectrum and frequency, consistent with Kolotkov, D. Y. et al. (2016). However, we follow Gilman et al. (1963) to define a relationship between the power spectrum and frequency. This incorporates the lag-1 auto-correlation of the timeseries into the theoretical red noise power spectrum (see below).

### B2.2    Theoretical red noise test

Alternatively, one can compute a theoretical power spectrum of the red noise (c.f. Gilman et al. 1963)

$$S(\nu) = \frac{1-r^2}{1-2r\cos 2\pi\nu + r^2} \tag{B7}$$

where $S$ is the power spectrum of red noise, $\nu = 1/t$ is frequency, and $r$ is again lag-1 auto-correlation from our data. For each frequency estimate we must multiply $S(\nu)$ by frequency range ($\Delta\nu$) (c.f. Eq. (B2)) to obtain a theoretical estimate of the (mean) energy of the red noise ($E^{red}$) (c.f. Kolotkov, D. Y. et al. 2016)

$$E^{red}(\nu) = S(\nu)(\nu\alpha - \nu/\beta) \tag{B8}$$

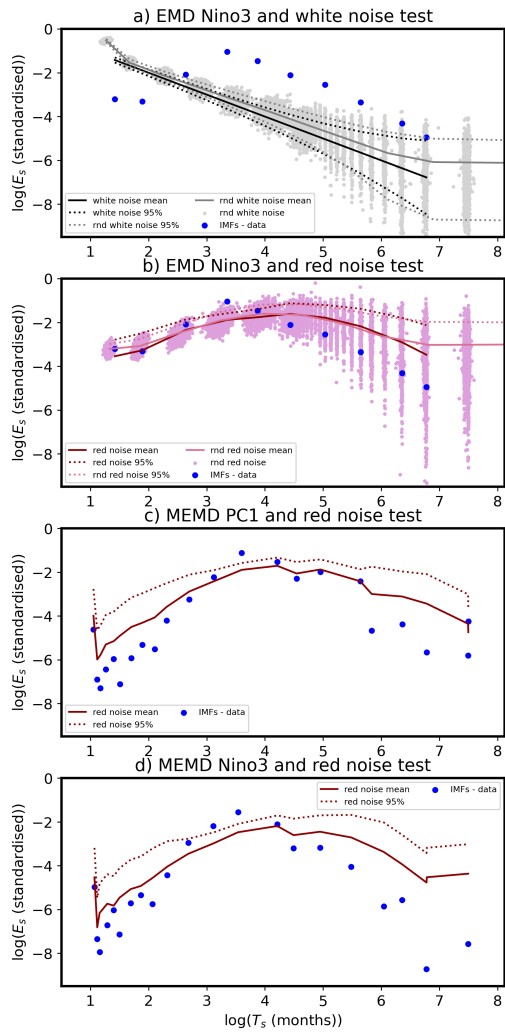

**Figure B1.** Significance tests for EMD and MEMD: (a) white noise significance test and (b) red noise significance test for EMD-IMFs of Niño3 index (blue dots); (c) (theoretical) red noise test for IMFs of PC1 of the combined field (via MEMD; blue dots) (d) (theoretical) red noise test for Niño3 obtained via MEMD (blue dots). (a) black solid line represents the theoretical linear relationship between the logarithms of period ($T_s$) and logarithm of energy density ($E_s$) (Eq. (B3)), black dotted line represents 5th-95th percentile (Eq. B5), respectively; grey dots represent $I = L$ realisations of IMFs of white noise timeseries (length is the same as for Niño3.4 index), whereas grey solid and dotted lines represent their mean and the 5th-95th percentile, respectively. (b) red solid line represents the theoretical red spectrum energy density (Eqs. (B7-B9)), red dotted line represents the 95th percentile (via $\chi^2$-test); light pink dots represent $I = L$ realisations of IMFs of red noise timeseries (Eq. B6; length $L$ is the same as for Niño3 index), whereas pink solid and dotted lines represent their mean and the 95th percentile, respectively. In (c) and (d) the red solid and dotted lines are as in (b) but for the respective data from MEMD. For further description of the figure see text.





where $\beta = \sqrt{\nu_s/\nu_{s+1}}$ and $\alpha = \sqrt{\nu_{s-1}/\nu_s}$ with $s$ running over frequencies (from higher to lower frequency). Note that since EMD is a dyadic filter (each lower frequency is a half of the previous one; e.g., Flandrin et al. 2004; Rehman and Mandic 2011) both $\alpha$ and $\beta$ typically take a value of $\sqrt{2}$ (consistent with, e.g., Kolotkov, D. Y. et al. 2016). Note that when mode-mixing is present (e.g., in this study it is generally present at higher frequencies) this is not necessarily true, hence the use of $\alpha$ and $\beta$ in Eq. (B8). Finally, $E_{red}$ must be scaled such that its total energy is the same as the total energy of our data (e.g., Madden and Julian 1971; Bretherton et al. 1999)

$$E_s^{red} = E_s^{red} \frac{\sum_{s=2}^{s_{max}-1} E_s}{\sum_{s=2}^{s_{max}-1} E_s^{red}} \tag{B9}$$

where $s_{max}$ is the number of IMFs (as above), and $s$ represents the $s^{th}$ IMF of frequency $\nu_s$, and $E_s$ was defined above (Eq. (B1)). Note that we scale $E_s^{red}$ from total energies of IMFs between $s = 2$ and $s = s_{max} - 1$ as the last IMF is typically a trend/residual and the first IMF does not necessarily follow the distribution correctly (but including the two usually does not significantly alter the results). $E_s^{red}$ is shown in Fig. B1b as red solid line.

This red noise test is typically used in climate science to determine the significance of power spectra peaks in our data (using $S(\nu)$ from Eq. (B7)), and it differs from the red noise test of Kolotkov, D. Y. et al. (2016) as it takes into consideration the lag-1 auto-correlation of the data. If the cosine function in the $S(\nu)$ (Eq. (B7)) is expanded into Taylor series ($\cos 2\pi\nu \approx 1 - (2\pi\nu)^2/2 + ..$) one can realise that for large $\nu$ (high frequencies) $S(\nu)$ indeed reduces to the spectrum $\gamma\nu^{-2}$ (with $\gamma$ a constant) suggested by Kolotkov, D. Y. et al. (2016); Franzke (2009). However, for low frequencies (small $\nu$) they do not agree well and ultimately $S(\nu)$ also becomes a constant (see Fig. B2 for comparison). Furthermore, as $S(\nu)$ depends on lag-1 auto-correlation ($r$) we can see from Eq. (B7) that for $r = 0$, $S(\nu) = 1$, i.e., it reduces to the power spectrum of the white noise. This means that this theoretical test can potentially be used for testing the significance of the data that corresponds to either white or red noise.

The significance of the IMFs from the input data is tested using $\chi^2$-test, where $s^{th}$ IMF's $\chi_s^2$ value for the (one-tailed) 95th percentile is computed from $\mathrm{DoF}_s = L_{eff} E_s$ degrees of freedom (instead of $L E_s$ as was the case for white noise, due to strong correlations between neighbouring data-points; Bretherton et al. 1999; Wu and Huang 2004; Kolotkov, D. Y. et al. 2016), where (Bretherton et al., 1999)

$$L_{eff} = \frac{1 - r^2}{1 + r^2} L. \tag{B10}$$

Then we multiply the expected red noise curve $E_s^{red}$ by $\chi_s^2/\mathrm{DoF}_s$ (e.g., Madden and Julian 1971; Bretherton et al. 1999) to ultimately obtain a threshold for $95^{th}$ percentile (red dotted line in Fig. B1b). Note that for $\mathrm{DoF}_s < 1$ we set $\mathrm{DoF}_s = 1$ (to avoid numerical issues). The IMFs derived from the data (blue dots in Fig. B1b) that exceed the red noise threshold (i.e., lie above the red/pink dotted line in Fig. B1b) are considered significant at $95^{th}$ percentile (one-tailed).

Fig. B1b shows that the two (synthetic and theoretical) red noise tests (for Niño3 index via EMD) are somewhat comparable and that the majority of the input (e.g., Niño3 index) data (blue dots) lies within the red noise range (i.e., within the pink-dots, and below the pink/red dotted line). However, we can identify one IMF (period $\sim$31 months or $\sim$2.5 years) that is above the red noise threshold and well within the typical ENSO timescale (2-8 years), suggesting quasi-periodic behaviour (oscillations).

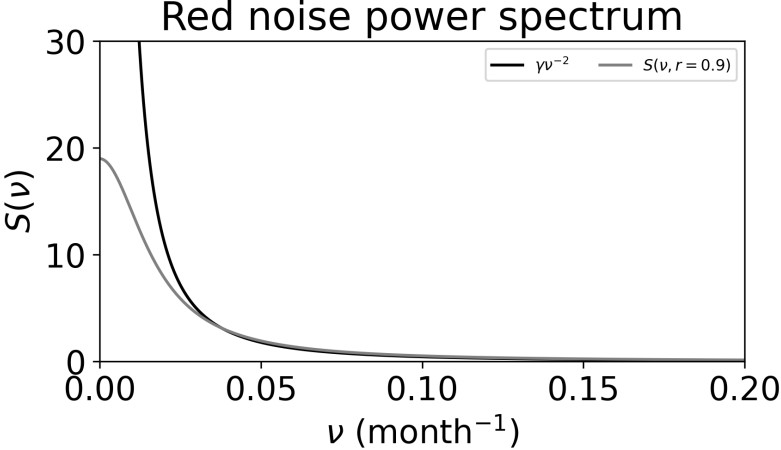

**Figure B2.** Red noise power spectrum ($S(\nu)$) for (black line) $S(\nu) = \gamma\nu^{-2}$, and (grey line) $S(\nu, r)$ from Eq. (B7) for $r = 0.9$. $\gamma$ was estimated as a ratio between the integrated power spectra of the two spectra for frequencies higher than 0.02/month ($\gamma = \sum_\nu \nu^{-2} / \sum_\nu S(\nu, r)$) where the two power spectra generally agree well.

Similarly, we can identify one significant IMF (via MEMD) in the PC1 (Fig. B1c) of the 4-variable field (discussed in section 2.1) with a period of ~37 months (~3 years). Note that we use PC1 of the combined field for identification of the
790 overall significant IMFs (via MEMD) here, since PC1 strongly dominates the EOF decomposition of the field. One could potentially also compute significant modes across all grid points, e.g., by averaging data over all grid-points, but we have not used this here. Instead, we use an additional test on map-plots in section 3 (Fig. 2), where we identify *potentially* "oscillatory" grid points and use grey shading on areas that are well represented with red noise alone (i.e., not significant).

Upon a reconstruction of IMFs for the x-y-t SSTs (Eq. A4), we can average SSTs over the Niño3 region (using the same
IMFs obtained via MEMD). This yields two significant modes (Fig. B1d) both within the quasi-biennial QB and LF/QQ range of the ENSO (Allan, 2000; Jajcay et al., 2018), ~23 and ~37 months (~2-3 years). This means that Nino3 index has 2 significant modes of variability (identified via MEMD), but the region of significance is narrow (small) for the ~23 month-mode, thus it is not apparent in the PC1 test that encompasses the whole tropical Pacific. Note that we do not necessarily expect the same results from EMD (Fig. B1b) and MEMD (Fig. B1d) methods since the MEMD finds a "synchronised" signal
within the tropical Pacific, whereas EMD only analyses the 1D Niño3 timeseries. Nonetheless, the two methods agree on the quasi-periodic timescale of 2-3 years in the Niño3 region. This is also consistent with the significant periods inferred from the usual power spectrum analysis of the Niño3 index (1D) obtained via Fourier Transform (Fig. B3; significant peak in black solid line (power spectrum of Niño3 index) is above the red dashed line (red noise one-tailed 95% threshold)), as well as via a 1D wavelet transform (not shown).



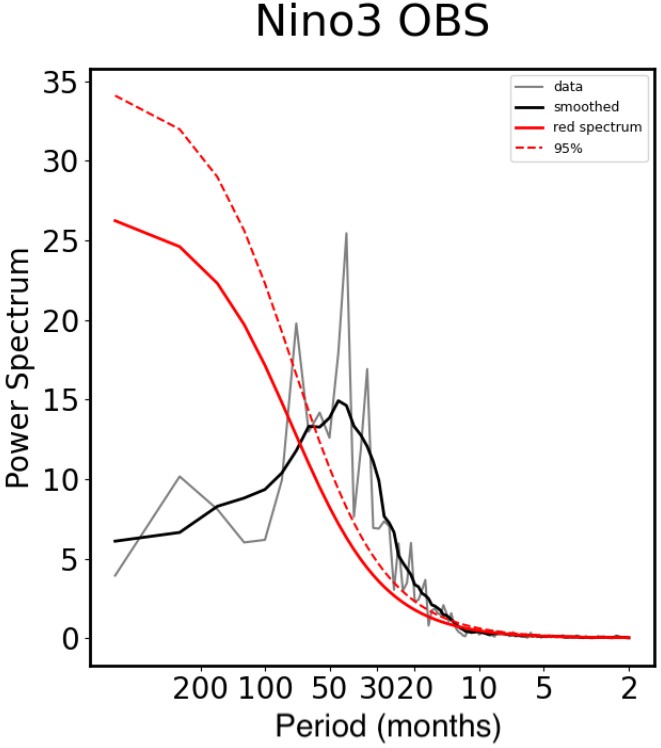

**Figure B3.** Power spectrum of Niño3 index. The power spectra are first computed for 500-months long chunks (overlapped by 250 months) and then averaged over all cases (grey solid line). The black solid line represents a 10-point running mean of the grey solid line (to increase the number of degrees of freedom, which is $f_\omega L/0.5 L_{\text{chunk}} = 10 \times 1680/250 \approx 67$; see also Boljka et al. 2021). The red solid and dashed lines represent the theoretical red noise test and its (one-tailed) $95^{th}$ percentile, respectively.

*Author contributions.* LB performed the analysis, prepared the figures, and wrote the first draft of the manuscript. NEO and NSK provided additional insight and helped improve the manuscript for the final version.

*Competing interests.* The authors declare no competing interests.

*Acknowledgements.* We thank Lander Crespo for helpful discussions, and Ingo Bethke for the help with the data. This work was supported by the Trond Mohn foundation (project BCPU, grant number BFS2018TMT01) and was performed on NIRD/Sigma2 (project NS9039K).



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
