# Peer review of "Identifying quasi-periodic variability using multivariate empirical mode decomposition: a case of the tropical Pacific"

_Weather and Climate Dynamics, 2022_

## Author Response (AR1)

**Responses to the Reviewers**

**Reviewer 1**

**Summary:** The authors use a multivariate empirical model decomposition (MEMD) method to find the dominant quasi-periodic variability in the tropical Pacific and find two modes with ~2 and ~3 year period, that can reproduce the bandpass filtered SSTa of Nino3 very well. The ENSO dynamics are further analyzed in conceptual models and a climate model and finally the prediction skill of a statistical model based on MEMD is shown.

**Overall opinion:** This study is an interesting study to an important topic of ENSO research and it is well structured and well written. The results are well elaborated and convincing. I have only a few major and minor comments that need be addressed before this manuscript is ready for publication.

We thank the reviewer for careful consideration of our manuscript and their constructive comments that have helped improve the original manuscript. Note that line numbers below in our responses refer to the revised manuscript.

**Major comments:**

Sect. 3.1 and Fig. 2: Why do you only show IMF13 in Fig. 2 and not IMF12? Maybe IMF12 represents the nonlinearity of ENSO as EOF-2 does? Please show the same analysis for IMF12 as shown for IMF13 in Fig. 2, at least as supplement so that one gets a feeling how these two modes interact with each other to represent ENSO.

We have now put IMF12 in the supplement (see Fig. S2) and referenced it in the main text (e.g., l. 328). The main reason for not showing IMF12 initially was because it looks virtually the same as IMF13 (as stated in the manuscript; see l. 252-253, 345-346) but has smaller amplitude and much smaller region of significance (now further emphasised; e.g., l. 854-858). The issue is that both modes are composited over the Nino3 index (virtually the same as EOF1/PC1), which is of interest in the present study. Thus, we pick up East-Pacific (EP) ENSO, not, e.g., Central-Pacific (CP) ENSO or EOF2/PC2 (i.e., we do not pick up different flavours of ENSO). However, if one looked at the temporal (spatial) evolution of IMF12/13 (i.e., not composite) we believe we could see some interesting (spatial) evolutions, since the method is non-stationary and should be able to identify different patterns for each ENSO event.

Sect. 4.1 Model evaluation: Please show same analysis as shown in Fig. 1 & 2 for your climate model so that one can see how well these modes represent ENSO in the climate model and to highlight similarities and differences!

The analysis has been added in the supplement (see Figs. S3-S5). We also provided an equivalent of Fig. 5 in Fig. S6. The main text has been amended accordingly (see l. 516-518, 520-521, 536) and further references to these Figs. have been added where relevant (see track changes). There are differences in the model that primarily stem from model biases (e.g., ENSO

SST warming extends further west; amplitude is too strong likely due to too strong periodicity; etc.), but overall it looks similar (ENSO-like).

**Minor comments:**

L47-58: "This feedback alone would result in continuous warming (cooling) in the eastern tropical Pacific, therefore negative feedbacks are necessary for quasi-oscillatory behaviour in the eastern tropical Pacific ..." I am missing here the discussion of the contribution of the seasonal cycle for the growing and decaying of ENSO events, as positive feedbacks are in general stronger in boreal autumn and the negative feedbacks are stronger in boreal spring (Wengel et al. 2018). Is this seasonal forcing included in these conceptual models? The seasonal cycle is a very important contributor of growth and decay of ENSO.

We agree that in general the seasonal cycle matters for phase-locking of ENSO. However, these oscillator models are very simple and do not (to our knowledge) include seasonal cycle, though one could imagine a more complex model of ENSO that includes it (e.g., Stein et al 2010). Alternatively, the seasonal cycle may still be present indirectly as it helps phase-locking the ENSO signal, which peaks in winter in data with the seasonal cycle removed as well.

In this study, we removed the seasonal cycle from the timeseries analysis and as such we do not really consider it (see l. 135-142). We further emphasise this on l. 142-144; we have also added a note on the importance of the seasonal cycle in the introduction (see l. 60-62).

L103 & L106: I do not understand why you state here "not shown" in the introduction. What don't you show? Please make clear or delete the "not shown".

Any reference to NAO, MSLP etc. has been removed (except for conclusions/future work).

Fig. 3: Please improve the legend in Fig 3, as it is hard to distinguish solid, dashed and doted lines in the legend.

We have updated Figs. 3,4,6 with clearer legends.

L302f: "and deeper thermocline in Niño6 region" In Fig. 2a the thermocline is deepest at the equator in the western Pacific and not north of the equator. Please change.

In this phase, thermocline reaches its deepest "phase" across the western Pacific (including Nino6). We have amended the text accordingly (see l. 336, 647).

Fig. 4 caption: Why do you give a range for the high-pass and low-pass filter? Normally you just give one cutoff period.

The reviewer is correct. Here, we merely take a range of higher and lower frequencies around the intermediate quasi-periodic band to show how relationships between variables can change. This means that we do not take a high/low-pass cut-off, but rather we use 2 additional band-pass filters of different (higher and lower) frequencies (consistent with IMF11 and IMF14).

We have clarified this in the caption of Fig. 4 and we used "higher"/"lower" frequency or "longer"/"shorter" period instead of "high"/"low" frequency (e.g., l. 444, 458).

L516f: "Similarly, predictions from tropical Pacific mean thermocline depth are not considered here as they did not improve predictions (not shown)" This sounds strange to me as the mean thermocline depth gives us the predictability of 6 months in climate model predictions. Please explain why you think that it reduces the prediction skill here!

L546f: Similar as above: Why is prediction with only SST better than with all variables in a)? This sounds strange to me. Please explain!

We address the two comments together: This was surprising to us as well. Perhaps it is because other variables can have other processes embedded in them - like deep-ocean dynamics (thermocline) or high frequency atmospheric processes (wind stress). These can then interfere with a statistical prediction of SSTs, which is computed on raw (unfiltered) data. It may also be that wind stress and thermocline are better represented as red noise even though SST (Nino3) is periodic, which may then act to weaken the predictability. Also, we use a very simple linear statistical model and as such it is rather hard to compare it to a dynamical model. Note, however, that the main point here is that smoothing of the predicted signal within the 2-4.5 year band yields good prediction-skill due to quasi-periodic nature of ENSO on those timescales (regardless of variables used).

We have added clarifications along those lines (see l. 589-596).

**References**

Wengel C, Latif M, Park W, et al (2018) Seasonal ENSO phase locking in the Kiel Climate Model: The importance of the equatorial cold sea surface temperature bias. Clim Dyn 50:901–919. https://doi.org/10.1007/s00382-017-3648-3

**Reviewer 2**

The study applies multivariate empirical mode decomposition (MEMD) to analyze four variables associated with ENSO. Two intrinsic mode functions (IMFs) derived from the MEMD have their characteristic time scales of about 3 years, that match with ENSO variability. The time series of IMFs indicates a quasi-periodic ENSO variability and is consistent with oscillators. The authors demonstrate a novel pathway to explore the intrinsic dynamics of ENSO, and I believe that the article could be published. However, I have some concerns about the details of the methodology and suggestions for writing.

We thank the reviewer for careful consideration of our manuscript and their constructive comments that have helped improve the original manuscript. Note that line numbers below in our responses refer to the revised manuscript.

**Major comments:**

1. The author clearly introduces the MEMD method in the appendix. However, in section 3, the author jumps to IMF12 and IMF13, and I am a little bit confused. Hence, I suggest inserting a section (or a section in the appendix) to summarize more details about all the IMFs. The authors should present the time series of all the IMFs, and enlist a table with characteristic frequencies for each IMF before the discussions of IMF12 and IMF13.

We have now plotted all IMFs of SST (Nino3), wind stress (Nino4,5), thermocline depth (Nino6, Pacific mean) - see Figs. S7 - S11 in the supplement. We have also provided all their characteristic timescales in Table S1. Note that we have put these figures/table in the supplement to avoid having too much information in the main text (manuscript is long as is). Fig. B1 actually shows typical timescales of each mode in a graphical way (it is timescale versus amplitude, but in a logarithmic scale). We have also realised that we never mentioned IMF12,13 in Methods and then we suddenly jumped to them in section 3 (as the reviewer pointed out).

We have added additional clarifications in the Methods section 2.2 - see l. 187-189, 232-233, 243-244, 245-252.

Also, the author needs to clarify more about Fig. B1. For example, given the doubling periodic of each consecutive IMF, it is reasonable to have 10 IMFs using the EMD for ~100-year monthly mean data (Fig B1a, B1b). However, why are there 22 IMFAs from the analysis of PC1 (Fig B1c) and 21 IMFAs from the Nino3 (Fig B1d)?

The number of IMFs for PC1 and Nino3 is the same, i.e., 22 IMFs, but one IMF in Nino3 (Fig. B1d) was too large and did not fit in the Figure. As shown in the supplement now, all PCs and all variables have 22 IMFs and IMFs have similar frequencies across different variables/PCs (see Figs. S7 - S11 and Tables S1, S2).

As for the number of IMFs, we believe that the number of IMFs is larger due to more timeseries (i.e., 20 PCs) input into MEMD algorithm versus one timeseries input in EMD algorithm. The

different timeseries can have slightly different timescales represented and MEMD can pick them up (see also answer to major comment 2 below). Note, however, that the number of IMFs can also change depending on the parameters used to run MEMD (e.g., "fix_h" parameter mentioned in the text).

We have now added discussions around these points in the caption of Fig. B1 and in the text as well (see l. 861-869). Fig. B1 is also further described on l. 769-776.

2. I also suggest that the authors could clarify more about the "multi-variable" parts of the EMDs. For example, it is well-known that the sea-surface temperature is generally more "smooth" than wind stress. It is more likely that the IMF1 from the wind stress data carries more high-frequency noises than the IMF1 from the SST. Hence, the IMF1 from the wind stress and SST might have different characteristic frequencies. Again, the authors could present some figures of raw data and IMFs to clarify if each IMF function from different variables have similar frequencies.

MEMD's job is to identify typical timescales within the system of input variables and provide IMFs that can describe timeseries of all these variables. This is now shown in Figs. S7-S11 and Tables S1, S2, where IMFs across all variables and PCs generally have similar timescales. Thus, IMF1 of wind stress ultimately has the same timescale as IMF1 of SSTs. However, because input variables (or grid points) have different timescales represented, e.g., in the high frequency range (wind stress is noisier than SSTs), this can cause mode mixing in MEMD analysis when no clear timescale can be identified across different timeseries. IMF13 (or an equivalent mode with similar frequency range) thus typically emerges across different parameters and from EMD or MEMD analysis. This is because it is such a prominent mode of variability in the tropical Pacific, but other timescales can be more problematic. Again, see l. 861-869.

3. One more concern about the IMFA forms by the addition of s'th IMFs of the 20 PCs(with EOFs). However, what if the s'th IMFs have different characteristic frequencies for different PCs? It is very likely that there is a big difference between the frequencies of IMF(s, m) and IMF(s, m+1). This issue might enhance the mode-mixing and further jeopardizes the analysis. Hence, I suggest presenting the EOFs/PCs results (in the appendix or before section 3) with at least a table of PCs. In this case, readers can judge how PC1 dominates so that we don't have to worry too much about the issue.

As mentioned above, MEMD is designed for identification of common timescales across input timeseries. Thus, IMFs of all PCs should have similar frequencies within the range of frequencies belonging to that IMF (i.e., 6.7th and 93.3rd percentiles) - see also Table S2. Mode mixing does, however, occur at high frequencies, where timescales are not clear (see comment 2 above for further explanation & l. 861-869).

We have also decided to use SST (Nino3) as index for compositing and all the analysis, not PC1 to avoid issues with the PC1 dominance (see l. 324-329, caption of Figs. 2,3). For the red noise test, one can use different timeseries as basis and PC1 was just our initial thought (its

analysis is kept for reference in Fig. B1, but not used further). We have realised that it is confusing and that using SST (Nino3) throughout the study makes everything more consistent across methods and figures. We updated numbers, figures and texts accordingly (see tracked changes and updated Figs.).

**Minor comments:**

1. (L126-129) I am a little confused about the "30-year running mean seasonal cycle". Do you take the 30-year climatology of seasonal cycle, and then make a moving window of this climatology? Need clarifications.

We have added more text about the 30-year running mean (see l. 134-144).

2. (L162) "Where timescales are not clear" I suggest giving a range of frequencies.

Added to l. 184.

3. (L197) "standard deviation" Needs to clarify how whether the standard of each variable or four variables.

Each variable in each grid point is divided by standard deviation separately - clarified on l. 223-224, 703.

4. (L232) Would IMF11 and IMF14 be involved with the ENSO? Need clarifications.

All IMFs together make up ENSO, but most of them follow the red noise spectrum. We find that the most important ones in our analysis are IMFs 11-15, which ultimately give most of the amplitude to ENSO, but only IMF13 and to some extent IMF12 are quasi-periodic. This was mentioned further down in the text, so we have now specifically mentioned IMF11,14, 15 on l. 305-308. Note (again) that we detrended timeseries, thus (multi)decadal modes of variability (like PDO, IPO) are not present in our analysis.

5. (Fig 3,4,6) The legend line styles are hard to discriminate; suggest rearranging the size.

We have fixed the legend in Figs. 3,4,6.

6. (L359) Need more details about why the time scale is about three years, or insert a reference.

We simply ran the conceptual oscillator model with provided equations, which yielded timeseries with a timescale of about 3 years. This was estimated through the timelapse between different maxima in timeseries (different "events"). We have moved up a paragraph about computing timeseries from Eqs. (5-8) and added the clarification about period. See l. 392-397.

7. (Fig 5) Hard to tell the line. I suggest using different colors or line widths.

We used these colors/lines as they are color-blind friendly (journal's policy). Also, they are consistent with lines in Fig. 3. We have now made dotted lines in Figs. 3-6 blue (similarly in

the supplement) as this line was the hardest to read. Note that this line is very closely related to the grey-solid line. Wider lines were not used as that makes it harder to see the time evolution of all variables (feels crammed). Thus, "black dotted" was changed to "blue dotted" "line" everywhere in the text as well – see track changes.

8. (L521) I suggest giving meow descriptions about the linear-regression models.

We have added a schematic that can hopefully explain how the data were lagged before they were input into the statistical prediction model. - see new Fig. C1 and its caption. We have also moved the details to the Appendix C to make the main text easier to read.

9. (L675) "filed" is a typo.

10. (L679) Replace "=" by "\simeq".

11. (L682) Same as (L679).

We have fixed the typo and "equality" symbols as suggested. See l. 720, 724, 728.

**Reviewer 3**

The manuscript "Identifying quasi-periodic variability using multivariate empirical mode decomposition: a case of the tropical Pacific" by Boljka et al. starts out with introducing a new statistical method to define modes of variability in the tropical Pacific that are related to ENSO. The method is non-linear and non-stationary and can therefore lead to potentially very different results to traditional linear and stationary methods. The manuscript then further gets into developing a new conceptual model for ENSO based on previous models and further gets into presenting a prediction model of ENSO based on the modes from the new statistical method. The topic addressed by this manuscript is interesting and potentially worth publishing. However, the manuscript is very hard to follow and many aspects are not well explained. It is difficult to see what the outcome of this work is and how the new method can help us to understand ENSO, rather then just being a fancy 'trick'. I cannot recommend to published the current version of this manuscript and would recommend a major revisions before it may get published. Detailed comments below.

We thank the reviewer for careful consideration of our manuscript and their constructive comments that have helped improve the original manuscript. Note that line numbers below in our responses refer to the revised manuscript.
* * *
**major points**

(1) The new method: The authors do try to explain the method, but unfortunately it fails to be clear enough. This would need to be improved. It would help greatly if the authors could more clearly explain what the elements of an MEMD mode are. Ideally in reference to something that most readers know (e.g. EOF-modes). An EOF-mode, for instance, has an eigenvector (pattern), eigenvalue (amplitude) and principle component (time series), they are ordered by explained variance, and are orthogonal to each other both in terms of eigenvectors and PCs. What are the elements of an MEMD mode?

The elements of an (M)EMD mode will ultimately depend on input data. In its most basic form, its elements (output data) would be timeseries for different timescales that can be identified within the data - so frequency/period/timescale can be another element. Modes are ordered from highest frequency (shortest period) to lowest frequency (longest period) and trend (the last mode is considered a trend of input timeseries). See l. 153-155, 229-230, 726-727, Appendix A, Figs. S7-S11 as examples for this from ENSO data, and Section S.3 for a simple example (as suggested by the reviewer in the comment below).

If spatio-temporal data are used, we first reduce their dimensionality via EOF/PC analysis (like in this study). Then, the input data for MEMD are, e.g., 20 PCs (each PC has an EOF pattern associated with them). The output from MEMD analysis are timeseries (IMFs) of these PCs, where each PC is split into a certain number of IMFs (here we have 22) of which each IMF has a specific timescale associated with it (and is consistent across the 20 PCs). This is now shown

in Table S2. For each IMF we can then take the 20 PCs and their EOFs and reconstruct spatial pattern evolution as described in Appendix A. Thus, we can also extract spatial evolution associated with each IMF (e.g., shown in Figs. 2, S2 for a composite over Nino3 index for IMF13,12, respectively), which is non-stationary (evolves with time; i.e., it is not just one constant/stationary pattern like in EOF analysis).

As mentioned above we have now provided additional text, plots and tables (as also requested by other reviewers) to help the readers understand MEMD better. For further visualisation of the (simpler) 1D method, the reviewer is referred to Wu and Huang (2004; doi: 10.1098/rspa.2003.1221) or this website.

Overall it appears that the method is a fancy trick (that is hard to understand) to time filter the data. What is the value? What do we learn from this method that we did not already know?

We believe that the main value of (M)EMD is that it can objectively identify intrinsic timescales within the timeseries (i.e., without predefining a range) and output timeseries for each timescale (IMFs) in a nonlinear and nonstationary way. It will also extract the co-varying part of the system, i.e. identify common timescales across different timeseries (see l. 147-157). In this study, we then use the output timeseries (IMFs) and their frequencies to extract periodic signals based on red-noise test (as described in Appendix B). This is a novel way for identifying periodic modes of variability using (M)EMD (hence the lengthy Appendix B), even though the red noise test is applied in a very similar manner as in a regular periodogram. Thus, we use MEMD to identify frequency range in which ENSO is quasi-periodic (roughly 2-4.5 years), without any predefined range (see l. 155-157) and with a method that is nonlinear/nonstationary (l. 147-149). Note that the frequency range is identified because of the non-stationarity of the method, i.e. timescale varies in time (see Fig. 1). With this method we are also able to identify regions of quasi-periodic variability (e.g., across the tropical Pacific in Fig. 2 – significant points, i.e., without grey shading). However, the main focus of this study are the ENSO oscillator models, for which we found some inconsistencies with the existing theories by filtering timeseries to the timescales that MEMD yielded (see section 3.2 in the manuscript). For example, IMF11 & IMF14 (still somewhat within ENSO timescale range of 2-8 years) exhibit very different dynamics from IMF12,13 (Fig. 4 vs. Fig. 3) and we only realised this once we plotted equivalent Figs. to Figs. 3,4 for different IMFs. Thus, the focus here is on oscillator models, and we keep most of the MEMD discussion in the Appendices/supplement (see also responses to comments below).

Note that we have successfully used this method (MEMD + red noise test as presented here) on other fields (not yet published) to identify regions that have potential quasi-periodic variability and have not been established before. Here, such a region is clearly identified across the tropical Pacific in Fig. 2 (as mentioned above) - though in this case the quasi-periodic region was known in advance. Thus, we believe there is value in using this method.

Often it is easy to illustrate this with an idealised constructed example that has strong similarity to the real world, but at the same time can be controlled, so that the 'truth' is known.

We have added an example with sine waves of different frequencies to show how the method works. Of course, in this case any linear/stationary method would be capable of identifying these simple modes - see Section S.3 in Supplement. This is merely to show that MEMD can extract correct waves (even if it is at times imperfect).

We also chose ENSO initially as a real-world example that is generally well-known. ENSO is a well studied phenomenon and thus a great starting point for understanding a new method that may be used for identifying quasi-periodic signals. However, we realised later that there were some inconsistencies with oscillator models, thus we focused on that here.

(2) Red noise test: It is not clear how this is done. Fig. B1 shows one blue dot above the 95% line, but is unclear what this blue dot is and how this would relate to the two MEMD modes IMF12 IMF13. How are the two modes selected and what does is has to do with Fig. B1?

The procedure goes as follows: (1) input timeseries of interest (e.g., 20 PCs) into MEMD algorithm; (2) get, e.g., 22 IMFs for each PC as output from MEMD; (3) for each IMF reconstruct data (from 20PCs and their EOFs) to yield spatial patterns of each IMF; (4) average data within, e.g., Nino3 box for each IMF to get Nino3 timeseries for each IMF (i.e., 22 timeseries of different timescales, 22 IMFs for Nino3); (5) compute average frequency/period as well as squared amplitude of each IMF; (6) get natural logarithm of both squared amplitude and period of each IMF; (7) plot log(period) vs. log(amplitude) of each IMF as scatter plot shown in Fig. B1d. So, in Fig. B1 each dot represents the mean amplitude and frequency of each IMF that we have identified within our timeseries. If we count dots from left to right we quickly realise that the 12th and 13th point (i.e., IMF12,13) are above red noise threshold (i.e., red dashed line) in Fig. B1d. Different panels are just for different timeseries/methods as described in caption (and text in Appendix B). We describe MEMD method in detail in Appendix A, i.e., points (1) to (4) above, whereas construction of red noise test and thus Fig. B1 is done (in detail) in Appendix B - see equations and their descriptions. We have added further clarifications on points (5)-(7) to l. 769-776.

The main idea for the red noise test comes from a "regular" periodogram. Since each IMF has an amplitude and frequency, we can imagine them as a periodogram, i.e., each IMF represents a different point on a periodogram. From that we can then use the "usual" red noise test (for periodograms - like in Fig. B3), which lets us find IMFs (blue dots) that are above red-noise threshold (above red/pink dotted line as stated in the Appendix's text - see l. 838-839). Please note that similar methods have been used in previous studies for EMD (as cited in Appendix B). Again, this is described in detail in Appendix B2 with equations. We also use the normal periodogram and its red noise test in Fig. B3 - merely to show that Fig. B1 yields IMFs with "correct" timescales as significant.

(3) Data from 1870 for thermocline depth and winds: It seems unlikely that much is gained in this analysis by using data going back way before we have any confidence in the observed data. Why is this done? Why does this analysis require such a long period of time using probably very poor data?

We do this mostly for consistency (and significance). When we constructed the prediction model we needed as much data as possible due to many lags used and the need for a "testing" and "training" set. Also, statistics are better. While not much is gained, also nothing is lost as the 19th century data does not really contaminate the timeseries analysis, but it can change the average timescale/amplitude of the timeseries (as its timescale/amplitude varies with time). This is not an issue as the modes that we find are consistent with modes identified in previous work using other methods and over shorter periods, e.g., 1970+ (as stated on l. 282-288). We initially also used OISST data (1980-present), and MEMD results were consistent with the presented results here (not shown).

We have added a few sentences addressing this on l. 111-114.

(4) Section 3.2 "Conceptual oscillator models of ENSO": This section is largely unclear and much of it appears to be unrelated to the topic. This needs substantial rethinking. HOw does it relate to the MEMD?

I recommend to strongly reduce this section.

The whole introduction/abstract/conclusions are focused on ENSO dynamics and ENSO's conceptual models. As this is the main focus of this study we have kept the sections the same. However, to avoid confusion about the main topic, we have changed the title to "Revisiting conceptual oscillator models for the quasi-periodic component of the El Nino Southern Oscillation".

MEMD here is used as a way of identifying a range of frequencies that yield quasi-periodic behaviour within ENSO (Nino3 region), i.e., it is merely used as a tool (but a lot of description was needed as it is a relatively new method). After that, we re-evaluate oscillator models within this particular range of frequencies (see also response to major comment (6) below and responses to major comment (1) above). We considered reducing the MEMD section in Methods instead, but then the rest of the text would likely be harder to follow.

We have added/removed a few sentences throughout the manuscript to keep this thread - see, e.g., l. 9-14, 78, 93-94, 103, 145, 218, 259-260, 316-317, 352, 496-512, 551-554, 610. See also track changes.

(5) Section 4.2 "Prediction": This section is hard to follow. It is not easy to understand what is predicted here. It seems to be a highly filter part of the NINO3 SST? Not clear what is learned from it. Highly filter SST is more predictable?

One of the goals of detecting and understanding quasi-periodic variability is to identify the predictable components of the climate system and isolate them from the (red) noise. With the prediction presented in section 4.2 we demonstrate that tropical SSTs in the frequency-band we identified as quasi-periodic via MEMD (i.e., IMF13) is indeed predictable well in advance (as expected) – i.e., IMF13 is the predictable component of the tropical SSTs. However, a wider frequency-band (e.g., IMF12 + IMF13) is less predictable.

Note that the statistical model is constructed on raw data (not filtered data), filtering is only performed as post-processing to see if the quasi-periodic band yields better prediction skill.

We have further clarified this on l. 551-554. Also, we have added a schematic on how prediction model is constructed in Appendix C and moved most of the text describing it there as well - hopefully this helps clarify section 4.2.

(6) last two sections: They seem to not clearly fit to the initial outline of this study, which is to introduce the new method. The connection is not that clear and it seem to drift into other aspects.

We apologise for the confusion. The initial idea was to introduce MEMD, but we later realised that ENSO oscillators are very interesting. However, we only realised the issues with these oscillators when we plotted line composites (cf., Fig. 3) for different IMFs. Thus, the two fit well together. As mentioned above, we have changed the title and tried making it clearer in the text that the main aim here is to use MEMD to re-assess oscillator models of ENSO. We have added comments throughout the text - see answers to comments above and track changes.
* * *
**Other points (in order as they appear in the text):**
* * *
line 101 "... ENSO teleconnections to the North Atlantic ...": The authors seem to indicate that they include SLP into the MEMD mode analysis to include teleconnections to the North Atlantic. This seems strange and is not further explored. It would also miss much more obvious teleconnections to many other regions on the globe. I would suggest to delete any reference to the NAO.

We agree with the reviewer and have removed such references except in conclusions (future work).
* * *
line 112 "This gives greater weight to SST data ...": Unclear! Later the authors state that the PC of the EOF-modes are used. So the data horizontal grid resolution should not matter. This needs clarification. Further, why would the SST be weighted higher? The later discussion focus on the ENSO dynamics including winds and thermocline. So, it should be equally weighted?

The horizontal grid resolution does matter. If more points come from SSTs then SSTs dominate the PC/EOF decomposition. If SSTs are not weighted more, then we can pick up internal ocean or noisy atmospheric dynamics from the other 2 fields that can interfere with the signal. For example, the domain we use for the MEMD analysis spans 25S to 25N in latitude (not just 5S to 5N used for Nino3/4 definition) to capture the off-equatorial thermocline depth that matters for the West-Pacific oscillator model. However, 25S/25N are further away from the Equator (0N) and can thus be affected by extratropical variability (i.e., not just tropical) – especially in the wind stress component, which can then in turn affect the intrinsic timescales present in the input data of MEMD analysis. Similarly, if the other 2 fields (wind stress & thermocline depth) do not have the same quasi-periodic variability as the SSTs, but instead are just red-noise-like, it can jeopardise identification of quasi-periodic variability of SSTs (via MEMD) when the 3 fields are combined. Thus, we suggest careful consideration of input data for MEMD. See l. 119-125.

Also, since the focus here is understanding Nino3/ENSO dynamics, we "enslave" wind stress and thermocline to the SSTs to extract variability associated with quasi-periodic nature of Nino3 index, which defines ENSO in the first place. Oscillator models were developed in the past for a better understanding of ENSO's quasi-periodic nature, which is primarily seen in the tropical SSTs. To remain consistent with previous work in trying to explain SSTs' quasi-periodic variability, we thus give more weight to the SSTs in our analysis.

Ultimately, the timeseries we get from quasi-periodic IMFs (12,13) can be extracted from band-pass filtered data as well. Thus, the results we obtain via our MEMD analysis are still representative of the variability in that frequency band not just for SSTs (Fig. 1) but also for wind stress and thermocline depth (not shown; implied via Fig. 3b).
* * *
line 116 "... whereas MSLP (not shown) is computed in the North Atlantic sector (80ºW - 30ºE, 25ºN - 70ºN).": Unclear why this is included. It seems unrelated to this work. This should be removed.

Yes, agreed.
* * *
line 135 "... methods methods ...": -> "... methods ..."

Done - see l. 153.
* * *
lines 139-145; description of the EMD method: Can this be visualised? What does the time series of the first mode look like? A figure may help.

Does this define a mode order? The first mode is the highest frequency and the last the lowest?

Yes it is exactly like this (see also above). See Figs. S7-S11 for IMF timeseries obtained via MEMD and Section S.3 for a simple example. Note that EMD is a well-established method and we just provided a quick recap. MEMD is much more complex and the reader is then referred to the relevant papers.

We have added additional descriptions/figure references, but slightly further down in the text - see l. 187-190, 229-230, Appendix A, and the answers to comments below/above.
* * *
lines 139 "(i) first we identify local minima and maxima of the timeseries ... "; description of the EMD method: I am already lost at step one. What time series? The data used are several fields, which have many time series. How does this work for MEMD modes?

We first introduce the EMD method (1D version) as it is much simpler to understand (this may not have been clear initially, so we added a few words on l. 159-160); then we extend it to MEMD, which is much more complex. In (i) we thus refer to "input timeseries" (added; see l. 160). MEMD solves a problem on n-spheres and is thus much harder to visualise or explain in a few sentences (see Rehman and Mandic 2010 for details; stated on l. 174-178).
* * *
line 232 "IMF12 and IMF13": What does "12" and "13" mean? How have these two been selected? Is it a coincident that they follow in order?

IMF12,13 are selected via a red noise test as described in Appendix B (Fig. B1). 12 and 13 mean 12th and 13th mode in terms of frequency ordering (see above as well).

It actually is a coincidence that we identify modes 12,13 - it could have been any other mode(s) as long as those modes are the ones that are quasi-periodic, which is a prerequisite for identification of significant modes in the present study. As mentioned above, we do not predefine a range in which periodic signals should occur.

Another prerequisite (sanity check) is for the modes to be well separated from other modes in terms of mean timescales (otherwise two modes may be "mixed" and thus not really separate). The latter is done somewhat subjectively based on Fig. B1 (e.g., Fig. B1c,d shows many dots at higher frequencies/lower periods that overlap or are very close to each other, but that is not the case around the significant modes, which are "well" separated). Note that the frequency ranges of different IMFs can be wide and overlap slightly on the tails of the "distribution" - see Fig. S1 for periodograms of IMFs12,13.

We have added additional notes along those lines on, e.g., l. 229-230, 244, 726-727, 852-860.

The authors find two modes: Does this mean two independent modes?

While EMD & MEMD are constructed so that modes would be largely independent, we can still find small correlations across different modes. This is due to non-stationarity of the method, which can result in limited covariability over limited time periods (Fig. 1 in the main text – l. 291-296). Here, we find a correlation of about 0.27 between IMF12 and IMF13 (as stated on l. 293). We added an additional sentence on l. 157-158 as well.

For instance, a periodogram may have 2-5 points in a series above the 95% confidence limit. Each point is an independent estimate of harmonic functions, but it does not mean that there are 2-5 "modes". So how should we interprete that there are two modes? Does it mean the time scales are broader?

Each IMF mode is associated with a range of timescales as mentioned in the text (see l. 266-276), i.e., IMF12 has a range 17-32 months and IMF13 has a range 25-52.5 months. Their mean periods are 22 and 34 months, respectively. See also Fig. S1 for periodograms of IMFs12,13. The range of timescales of each mode stems from the non-stationarity of the method and can be nicely seen in Fig. 1, where in some decades the variability in IMF12,13 is slower than in other decades (see also text around Fig. 1).

However, IMFs are timeseries and a direct output from MEMD, thus they are identified as modes of variability from MEMD by definition, but (as mentioned above) they can represent a range of timescales. Periodogram is a different method, but can be used to identify specific frequencies where quasi-periodic behaviour exists, which then allows us to filter the data to that specific frequency band with quasi-periodic behaviour. In the case of periodogram, we might then mix the two modes identified via MEMD into one "mode" as it is hard to identify separate modes of variability in a periodogram (see Fig. B3).

Please note that QB and QQ ENSO modes have been identified in the past, thus IMF12,13 are in fact "modes" as they strongly resemble those modes (see l. 282-288).

We hope that changes we have made to the text through this & other comments of the reviewer address also this comment.
* * *
lines 234-242 discussion of modes IMF12 and IMF13: This description of the time scale behaviour of the modes is not clear. Does it make sense to show the power spectrum of these two modes in comparison to the NINO3.4 SST power spectrum? I assume they both would have strong peaks at the peak in NIINO3.4 SST but have much less variance at all other frequencies?

This is exactly the case. IMF13 has a peak at about 34 months and IMF12 has a peak at about 22 months, and otherwise their spectra fall sharply away from peak frequency/period. See Fig. S1 (supplement) for periodograms of IMF12,13 and compare it to Nino3 periodogram in Fig. B3. We have also mentioned it in the text - see, e.g., l. 270, 291.
* * *
line 240 "... (e.g., in Fig. 3b), ...": It is unclear how Fig. 3b relates to what is said in this sentence. Wrong Figure reference?

Figure reference is correct, but the wording was clumsy. We have changed parentheses to "(e.g., 25-52.5 months band of IMF13 is used to construct band-pass filtered index composites in Fig. 3b; more below)" - see l. 274. Fig. 3b shows a composite for the same fields as Fig 3a, but uses the band pass filtered data with a frequency range consistent with IMF13. This is done for comparison of MEMD with some other method.
* * *
Analysis of phase: Why do the authors use 12 phases? A cycle is often illustrated by 4 phases (0,90,180,270 degrees). 12 seem to be too much. It would be better to use 0-360 degrees.

In Fig.3 and others: How is phase zero chosen? Often phase zero referees to the peak of the reference time series, but here it seems SST is the reference and it is negative at phase 0.

We use 12 phases to construct line-phase-composite plots, i.e., for smoothing/continuation purposes. Figs. 3,4,6 would look weird with just 4 points so we added 2 additional points between all main phases (also in Fig. 2 for consistency). Phases are chosen as: 0-30 (phase 0), 30-60 (phase 1), …, 330-360 (phase 11). This has now been clarified - see l. 325-329.

Also, we have now changed composites so that phase 0 is the peak of the composite. Before it was centred at -180 degrees. We have changed references to phases throughout the text accordingly - see track changes.
* * *
line 288 "... phase of the PC1 (of IMF13) timeseries ...": Unclear what a PC1 in this context means. PC1 is part of the EOF-modes?

Yes, PC1 comes from EOF analysis described in Appendix A and later used in Appendix B. However, to avoid confusion we have decided to change figures and text so that all composites are based on Nino3 index - for consistency. Again, see, e.g., l. 325-329, and Figure 2 caption.
* * *
Fig. 3 legend: The legend is not clear. There are several black lines and grey lines. Dashed or dotted cannot be seen in the legend.

Yes, we have fixed Figs. 3,4,6.
* * *
lines 491-492 "This suggests an issue ..., since the recharge-discharge mechanism, ... is correctly represented": Logic here is unclear. Where is the issue if it is correct?

Thank you for pointing this out. The issue here refers to the western Pacific, which seems to have erroneous variability. This is a known model bias/issue. Here, we see it mostly via different phasing of the variables in a phase composite in the model (Fig. 6) relative to observations (Fig. 3a). The eastern Pacific is relatively well represented, although it is too well tuned to the recharge-discharge mechanism (again, a known model bias). We have revised the text accordingly (see l. 528-546). See also Figs. S3-S4 for spatial composites in the model (e.g., warm anomalies during El Nino extend too far west and anomalies are "too" strong).

---

## Editor Decision (ED1)

At this point I cannot approve this manuscript for publication. Although the reviewers do not question the validity of the science, the third reviewer indicates that the presentation lacks clarity and leaves a reader confused. As my editorial colleagues have already pointed out, this paper aims to present a lot of material in a very short space which, while possible, requires careful crafting of the arguments to guide a reader through. Shortening the paper by putting bits of it into an appendix and changing the title does nothing to increase its clarity.

All the authors need to sit down together to discuss how best to present this work, as it currently falls short of the quality that's needed to be published in WCD. What is needed to move this paper towards being publishable, is a clear motivation for why the paper is long and then clear signposting as the paper unfolds to keep a reader engaged. Ultimately you need to convince a reader that it's worth reading this paper, otherwise no one will bother.

I suggest that the authors re-read and reflect on the original comments from the third reviewer, particularly those major comments where the reviewer requires more clarity. Although you did respond to these comments, the reviewer found the revision weak. As editor I agree: in reading the revised manuscript I had the exact same questions as the reviewer, thus the reviewer's comments remain unaddressed.

A revision to this manuscript requires edits that are more comprehensive than tweaking a few sentences. Reviewer 3's major comments should be more substantially addressed along with the new comments in their report. More clear motivation for what the paper hopes to achieve must be given. Sections should be restructured to give a clearer map for where the paper is going and how it will address the research question.

Below are some suggestions for how to increase the clarity, along with a few "Other Points" that I noticed while reading the manuscript.

**Motivation**

The foundations for the paper need to be laid out in the Introduction. By the end of the Introduction a reader needs to have had it clearly explained:

- Why do we need to revisit the various oscillator models for ENSO?
- Why do we need to use MEMD to understand the oscillators? What's wrong with existing methods?
- Why are you going to analyse both observations and output from a climate model?
- How does prediction fit into this whole picture (or drop this section)?

The answers to these questions may already be somewhere in the Introduction. However, as an independent reader coming to this manuscript afresh, I can assure the authors that it is currently far from clear what the answers are.

**MEMD**

The description of the MEMD is a very important part of this paper and deserves its own section. The authors may wish to de-emphasise the MEMD procedure but it is a critical part of the whole paper. If a reader doesn't understand what constitutes the components for the new oscillator, they aren't going to buy into the argument that a new oscillator is even needed – see comments from Reviewer 3.

Ending this section with a summary of the crucial points needed to understand MEMD would serve a reader well as a reference for use when reading the rest of the paper.

There are elements of the MEMD section that can't just be put into an appendix. The results in Appendix B2 are absolutely fundamental to the paper so need to be in the main text. Appendices present information that is additional to paper. If they were to be removed, the paper should still make sense: this is not the case for some of the appendices in this paper.

**Introduction**

I would suggest that the first 3 paragraphs of the Introduction are redundant: the audience for this paper already knows what ENSO is and what its impacts are. Getting straight to the point in this paper would shorten it and keep the reader engaged.

**Sections**

I would suggest that the sections are restructured somewhat to have more major sections. E.g. Intro, Data, MEMD, ENSO, Oscillators, Implications, Conclusions. Remember that it is highly unlikely that this paper will be read in one go , so breaking up the paper by theme will assist a reader when they return to it. Summaries at the end of each section will help the signposting for a reader. Similarly, high level overviews of what will be achieved in each section will help a reader understand what's going on. For example, Section S3 goes straight into the mathematical detail describing the method with no introduction to what will be achieved. It is better to give an overview first and then give the details. E.g. in section S3 begin with . "As an example we shall describe how MEMD is carried out on a simple periodic time series. We shall define 4 timeseries which have a shared angular frequency of pi/2 with other harmonics or phase shifts added on top. MEMD will be applied and should isolate this shared pi/2 mode. …"

There is much truth in the old mantra: "tell them what you are going to tell them, tell them, then tell them what you told them"

**Prediction**

I would suggest that this section be dropped. What purpose does it serve in addressing the subject: "Revisiting conceptual oscillator models for the quasi-periodic component of the El Niño Southern Oscillation"?

**Other points**

Paragraph beginning 115 – it is not clear from this paragraph why non-SST data are included. The emphasis of this paragraph is that SST is pre-eminent – SST weighted more in procedure, tau_x/thermocline are slaves to SST – so why even include the other variables?

Para beginning 130 – this is confusing, see comment from Rev3, it seems to imply that the variables are averaged into these boxes before they are fed into the MEMD. State what the purpose of this averaging is.

Line 155, using () here is very confusing. Just state " ordered from the highest to the lowest frequency, equivalently from the shortest to the longest period/timescale." () should only be used to indicate parenthetical statements, I.e. less important or explanatory statements, as per the usage in the following sentence. There's a weird precedent in some geosciences for their usage in other contexts, but it really should stop.

Section S3:

EMD description – in S3 add in an example of how the "envelopes" are computed in EMD to help the reader. This would help readers with a more graphic intuition.

It is Hard to parse this section.

State up front what you are doing at high level then dig into details, later. This applies to the whole paper. In most sections the procedure is described as it goes along so a reader has to work out what's going on while the detail is being described. It is better to give an overview first and then give the details. E.g. in this section state. "As an example we shall describe how MEMD is carried out on a simple periodic time series. We shall define 4 timeseries which have a shared angular frequency of pi/2 with other harmonics or phase shifts added on top. MEMD will be applied and should isolate this shared pi/2 mode. …"

Para line 196. This is a really important paragraph as it gives the motivation for the study. It needs to come way sooner in the paper.

Para line 220 – pull this out from the text for readers to refer back to. Present is as a list of steps.

Line 234 – surely this is the index of the IMFs not the IMFs of the index? These aren't the same.

Figure 1 – be clear that the time series are of nino3 of the IMF, not the IMFs themselves.

Confusion

Line 349 -

"Since we observe clear relationships between the relevant variables (e.g., Fig. 2) that strongly resemble a unified oscillator of Wang (2001a) (see also Fig. 5 in Wang 2018), recharge-discharge oscillator (e.g., Burgers et al., 2005), and others, we now revisit the theory of ENSO dynamics using the relevant conceptual oscillator models."

line 379 -

"However, the results from section 3.1 suggest that the average evolution on 2-3 year (average) timescale is different from the
 unified model and many other oscillator models discussed in Wang (2001a)."

This highlights the confused motivation for using the MEMD to revisit the oscillators.

---

## Author Response (AR2)

**Responses to reviewers and editor (W. Roberts)**

First, we would like to thank reviewers/editorial team for their patience and constructive comments on our paper. Below we first describe the latest changes to the manuscript and later provide point-by-point responses to the editor (W. Roberts).

Following the comments by reviewer 3 and W. Roberts we made substantial changes to the manuscript.

1. We have removed sections on oscillator models, results from a climate model (NorCPM1), and prediction implications. While the latter two have now been completely omitted, we still discuss implications for oscillator models in section 6 of the latest manuscript version, but we do not go into details. If the reviewers would like to have more text on oscillator models from the original manuscript, we could potentially add them in the supplement, though we deem it unnecessary.
2. We have re-ordered the manuscript so that MEMD algorithm and red noise significance tests are now described in more detail in the main text (sections 3 and 4 in the latest version). Similarly, we start the abstract, introduction, conclusion with statistical methods for climate science and MEMD, while ENSO is only briefly mentioned in the introduction and further expanded on in sections 5,6 of the latest version.
3. We tried emphasizing the novel results more.
   a. MEMD has not been used in climate science before (except for an idealized study), even though it is data-adaptive and thus fully nonlinear and nonstationary. The latter means that modes that emerge from MEMD, i.e., IMFs are nonstationary and thus have time-evolving spatial patterns of variability. Additionally, MEMD can identify timescales of variability within a given system objectively (without window pre-selection).
   b. We developed a red noise test, which has not been used with EMD or MEMD in the past. Now, both methods can be used for extracting quasi-periodic signals within climate system.
   c. We test MEMD + red noise test on tropical Pacific atmosphere-ocean variability as we know ENSO exists there and it thus provides a clear test case for the new method. We confirm MEMD can extract ENSO. Moreover, it can extract quasi-biennial and quasi-quadrennial ENSO modes as well as ENSO nonstationarity.
   d. MEMD modes that emerge are physical as they can isolate recharge-discharge oscillator and other relevant dynamics. We also find some interesting novel results for ENSO dynamics (e.g., different behaviour on quasi-periodic timescales versus other ENSO-relevant timescales) and suggest a re-evaluation of the unified oscillator model.

Additionally, we found an error in our PC/EOF analysis (inputs for MEMD) and hence some numbers/figures are now slightly different from our previous manuscript version. However, results remain consistent.

Hopefully, the manuscript is now clearer. Below are point-by-point responses to W. Roberts.

At this point I cannot approve this manuscript for publication. Although the reviewers do not question the validity of the science, the third reviewer indicates that the presentation lacks clarity and leaves a reader confused. As my editorial colleagues have already pointed out, this paper aims to present a lot of material in a very short space which, while possible, requires careful crafting of the arguments to guide a reader through. Shortening the paper by putting bits of it into an appendix and changing the title does nothing to increase its clarity.

All the authors need to sit down together to discuss how best to present this work, as it currently falls short of the quality that's needed to be published in WCD. What is needed to move this paper towards being publishable, is a clear motivation for why the paper is long and then clear signposting as the paper unfolds to keep a reader engaged. Ultimately you need to convince a reader that it's worth reading this paper, otherwise no one will bother.

I suggest that the authors re-read and reflect on the original comments from the third reviewer, particularly those major comments where the reviewer requires more clarity. Although you did respond to these comments, the reviewer found the revision weak. As editor I agree: in reading the revised manuscript I had the exact same questions as the reviewer, thus the reviewer's comments remain unaddressed.

A revision to this manuscript requires edits that are more comprehensive than tweaking a few sentences. Reviewer 3's major comments should be more substantially addressed along with the new comments in their report. More clear motivation for what the paper hopes to achieve must be given. Sections should be restructured to give a clearer map for where the paper is going and how it will address the research question.

Below are some suggestions for how to increase the clarity, along with a few "Other Points" that I noticed while reading the manuscript.

Again, thank you for these constructive comments.

**Motivation**

The foundations for the paper need to be laid out in the Introduction. By the end of the Introduction a reader needs to have had it clearly explained:

- Why do we need to revisit the various oscillator models for ENSO?
- Why do we need to use MEMD to understand the oscillators? What's wrong with existing methods?
- Why are you going to analyse both observations and output from a climate model?
- How does prediction fit into this whole picture (or drop this section)?

The answers to these questions may already be somewhere in the Introduction. However, as an independent reader coming to this manuscript afresh, I can assure the authors that it is currently far from clear what the answers are.

As mentioned above we have removed climate model & prediction results completely. Similarly, oscillator models are only discussed when considering ENSO dynamics, which is primarily done to ensure MEMD modes (IMFs) have physical meaning. However, we still find interesting novel results when we analyse variability of tropical Pacific on different timescales, and we show the nonstationarity of IMFs and ENSO.

**MEMD**

**The description of the MEMD is a very important part of this paper and deserves its own section. The authors may wish to de-emphasise the MEMD procedure but it is a critical part of the whole paper. If a reader doesn't understand what constitutes the components for the new oscillator, they aren't going to buy into the argument that a new oscillator is even needed – see comments from Reviewer 3.**

**Ending this section with a summary of the crucial points needed to understand MEMD would serve a reader well as a reference for use when reading the rest of the paper.**

**There are elements of the MEMD section that can't just be put into an appendix. The results in Appendix B2 are absolutely fundamental to the paper so need to be in the main text. Appendices present information that is additional to paper. If they were to be removed, the paper should still make sense: this is not the case for some of the appendices in this paper.**

We have put MEMD into a new section (now section 3) and slightly expanded it and added a figure to help visualise EMD (Fig. 1). Similarly, we have moved parts of Appendix B to the main text (section 4) and split the original Fig. B1 into 3 Figures – now Fig. 2 (relevant for MEMD and Niño3 index), Fig. S2 (relevant for MEMD and PC1 of the tropical Pacific), and Fig. B1 (relevant for the white and red noise tests discussed in Appendix B; relevant for EMD analysis only). Hopefully the methodology is now clearer.

**Introduction**

**I would suggest that the first 3 paragraphs of the Introduction are redundant: the audience for this paper already knows what ENSO is and what its impacts are. Getting straight to the point in this paper would shorten it and keep the reader engaged.**

We have removed most of the ENSO text from the introduction. We only keep text on red noise in the climate system on l. 43-50, and basic text on what ENSO is on l. 59-66. The rest of the ENSO discussion appears in sections 5,6. The introduction has also been shortened.

**Sections**

**I would suggest that the sections are restructured somewhat to have more major sections. E.g. Intro, Data, MEMD, ENSO, Oscillators, Implications, Conclusions. Remember that it is highly unlikely that this paper will be read in one go , so breaking up the paper by theme will assist a reader when they return to it. Summaries at the end of each section will help the signposting for a reader. Similarly, high level overviews of what will be achieved in each**

**section will help a reader understand what's going on. For example, Section S3 goes straight into the mathematical detail describing the method with no introduction to what will be achieved. It is better to give an overview first and then give the details. E.g. in section S3 begin with . "As an example we shall describe how MEMD is carried out on a simple periodic time series. We shall define 4 timeseries which have a shared angular frequency of pi/2 with other harmonics or phase shifts added on top. MEMD will be applied and should isolate this shared pi/2 mode. ..."**

**There is much truth in the old mantra: "tell them what you are going to tell them, tell them, then tell them what you told them"**

We have now split sections as suggested, though due to different content in the second half of the manuscript the sections are called differently. Namely, we now have sections: Introduction, Data, MEMD, Statistical significance test for climate, Tropical Pacific modes of variability, ENSO dynamics, and conclusions. Hopefully this breaks the paper down into smaller bits of information.

we have generally added some text at the beginning / end of sections to help the text flow, i.e., what is the goal of the next section or what will be or has been achieved in this section. For example, see l. 232-237, 270-272, 301-305, 346-348, 350-364, 427-431, 446-447. We have also added text to the beginning of section S3 in the supplement along the lines of the editor's suggestion. See l. 4-9 in the supplement.

**Prediction**

**I would suggest that this section be dropped. What purpose does it serve in addressing the subject: "Revisiting conceptual oscillator models for the quasi-periodic component of the El Niño Southern Oscillation"?**

This section has been removed.

**Other points**

**Paragraph beginning 115 – it is not clear from this paragraph why non-SST data are included. The emphasis of this paragraph is that SST is pre-eminent – SST weighted more in procedure, tau_x/thermocline are slaves to SST – so why even include the other variables?**

This is done primarily for testing physical meaning of modes of variability that MEMD yields. Ultimately, we find some interesting relationships as well (section 6). We have added more text on l. 75-77, 85-97.

**Para beginning 130 – this is confusing, see comment from Rev3, it seems to imply that the variables are averaged into these boxes before they are fed into the MEMD. State what the purpose of this averaging is.**

Averaging is performed after MEMD analysis. We added some text/equations to clarify this. First on l. 100-101, and later on l. 195-200. See also Eq. (1) for IMF spatio-temporal reconstructions.

**Line 155, using () here is very confusing. Just state " ordered from the highest to the lowest frequency, equivalently from the shortest to the longest period/timescale." () should only be used to indicate parenthetical statements, I.e. less important or explanatory statements, as per the usage in the following sentence. There's a weird precedent in some geosciences for their usage in other contexts, but it really should stop.**

On l. 137-138 we now use "The modes are automatically ordered from highest/shortest (mode 1) to lowest/longest (last mode) frequency/period." We also changed some parentheses use in other aspects, e.g., "SST (Niño3)" is now "eastern Pacific SST (Niño3)" to include less important information in the parentheses.

**Section S3: EMD description – in S3 add in an example of how the "envelopes" are computed in EMD to help the reader. This would help readers with a more graphic intuition.**

We have added envelope schematic to the main text (Fig. 1).

**It is Hard to parse this section.**

**State up front what you are doing at high level then dig into details, later. This applies to the whole paper. In most sections the procedure is described as it goes along so a reader has to work out what's going on while the detail is being described. It is better to give an overview first and then give the details. E.g. in this section state. "As an example we shall describe how MEMD is carried out on a simple periodic time series. We shall define 4 timeseries which have a shared angular frequency of pi/2 with other harmonics or phase shifts added on top. MEMD will be applied and should isolate this shared pi/2 mode. ..."**

We have added text along these lines on l. 4-9 in the supplement.

**Para line 196. This is a really important paragraph as it gives the motivation for the study. It needs to come way sooner in the paper.**

This has been moved to the introduction, see l. 36-43.

**Para line 220 – pull this out from the text for readers to refer back to. Present is as a list of steps.**

We have now made a list of steps. See l. 173-183.

**Line 234 – surely this is the index of the IMFs not the IMFs of the index? These aren't the same.**

Technically the index is split into the IMFs, but is computed on spatio-temporal IMFs. We now use "an index for each IMF (e.g., $IMF_s$ (SST (Niño3)) with s IMF-number)". See l. 197-198.

**Figure 1 – be clear that the time series are of nino3 of the IMF, not the IMFs themselves. Confusion**

We state that we have "timeseries of Niño3 index from IMF11, IMF12". This is now Fig. 3.

**Line 349 -**

**"Since we observe clear relationships between the relevant variables (e.g., Fig. 2) that strongly resemble a unified oscillator of Wang (2001a) (see also Fig. 5 in Wang 2018), recharge-discharge oscillator (e.g., Burgers et al., 2005), and others, we now revisit the theory of ENSO dynamics using the relevant conceptual oscillator models."**

**line 379 -**

**"However, the results from section 3.1 suggest that the average evolution on 2-3 year (average) timescale is different from the unified model and many other oscillator models discussed in Wang (2001a)."**

**This highlights the confused motivation for using the MEMD to revisit the oscillators.**

We have now omitted oscillators and the only motivation to use variables relevant for oscillators is to test ENSO dynamics in MEMD modes of variability. ENSO dynamics in its simplest form is related to conceptual ENSO oscillators. Additionally, as stated above we find some novel results. Hopefully text is clearer now. See the new section 6.

---

## Author Response (AR3)

**Response to Editor's comments**

EDITOR's comment – minor revision

Public justification (visible to the public if the article is accepted and published)**:**
I'm happy to accept this manuscript for publication subject to a few minor corrections. Both reviewers 1 and 2 have already accepted this manuscript. However, reviewer 3 felt that the changes made after the first review made the manuscript less clear than it was originally. It introduced a number of sections that did not add to the paper's main point and did not clarify areas that required clarification.
This revised manuscript removes the added sections and clarifies the points that were not clear in either the original or first revised manuscript, in line with the comments of reviewer 3.

**We thank the editor for careful consideration of our manuscript and for constructive comments throughout.**

The points that still require attention are:

Please clarify the section on the red noise test. In particular the paragraph beginning line 206 contains 3 possible ways of testing for significance. Please explain the situations in which each of the methods is the best choice.

**We have added some examples, though it is not limited to those. See l. 206-213. The choice is ultimately up to the scientist as they know what they are interested in. For the sake of robustness, it may be better to test different methods on the same data nonetheless.**

The sentence beginning line 208, "Thus significance of modes..." is not clear.

**This sentence has now been removed as it is unnecessary.**

Please check that all of the colourbars in figures have units, e,g, 4/5. If standardised state this on the colourbar.

**We have now stated units or mentioned that values are standardised next to the colourbar (figs. 4, 5, S3).**